# Tissue-specific mitochondrial HIGD1C promotes oxygen sensitivity in carotid body chemoreceptors

Alba Timón-Gómez[1†], Alexandra L Scharr[2†], Nicholas Y Wong[2†], Erwin Ni[2], Arijit Roy[3,4,5], Min Liu[2], Julisia Chau[2], Jack L Lampert[2], Homza Hireed[2], Noah S Kim[2], Masood Jan[2], Alexander R Gupta[6,7], Ryan W Day[6], James M Gardner[6,7], Richard JA Wilson[3,4,5], Antoni Barrientos[1*], Andy J Chang[2*]

[1]Department of Neurology, University of Miami, Miami, United States; [2]Department of Physiology and Cardiovascular Research Institute, University of California, San Francisco, San Francisco, United States; [3]Department of Physiology and Pharmacology, University of Calgary, Calgary, Canada; [4]Hotchkiss Brain Institute, University of Calgary, Calgary, Canada; [5]Alberta Children's Hospital Research Institute, University of Calgary, Calgary, Canada; [6]Department of Surgery, University of California, San Francisco, San Francisco, United States; [7]Diabetes Center, University of California, San Francisco, San Francisco, United States

*For correspondence:
abarrientos@med.miami.edu
(AB);
Andy.Chang@ucsf.edu (AJC)

†These authors contributed
equally to this work

Competing interest: The authors
declare that no competing
interests exist.

Reviewing Editor: David J
Paterson, University of Oxford,
United Kingdom

**Abstract** Mammalian carotid body arterial chemoreceptors function as an early warning system for hypoxia, triggering acute life-saving arousal and cardiorespiratory reflexes. To serve this role, carotid body glomus cells are highly sensitive to decreases in oxygen availability. While the mitochondria and plasma membrane signaling proteins have been implicated in oxygen sensing by glomus cells, the mechanism underlying their mitochondrial sensitivity to hypoxia compared to other cells is unknown. Here, we identify HIGD1C, a novel hypoxia-inducible gene domain factor isoform, as an electron transport chain complex IV-interacting protein that is almost exclusively expressed in the carotid body and is therefore not generally necessary for mitochondrial function. Importantly, HIGD1C is required for carotid body oxygen sensing and enhances complex IV sensitivity to hypoxia. Thus, we propose that HIGD1C promotes exquisite oxygen sensing by the carotid body, illustrating how specialized mitochondria can be used as sentinels of metabolic stress to elicit essential adaptive behaviors.

## Editor's evaluation

The arterial chemoreceptors are the body's primary defense against hypoxia. In particular, the carotid body glomus cells (type 1) are highly sensitive to decreases in oxygen availability where the mitochondria and plasma membrane signaling proteins have been implicated in oxygen sensing by type 1 cells. Here, Chang and colleagues identified HIGD1C, a novel hypoxia-inducible gene domain factor isoform, is essential for carotid body oxygen sensing, where it enhances complex IV sensitivity to hypoxia. Discovery of this protein and its function brings back into focus the importance of how specialized mitochondria can act as sensors to metabolic stresses like hypoxia.

## Introduction

The carotid bodies (CBs), located at the bifurcation of the common carotid arteries, are the major chemoreceptor for blood oxygen in mammals (*De Castro, 1928*; *Heymans et al., 1930*). Within

seconds of exposure to hypoxia (reduction in PaO$_2$ from 100 mmHg to below 80 mmHg), CB glomus cells signal to afferent nerves projecting to the brainstem to stimulate acute cardiorespiratory and/or arousal reflexes (*Black et al., 1971*; *Lahiri and DeLaney, 1975a*; *Lahiri and DeLaney, 1975b*; *Neil and O'Regan, 1971*; *Verna et al., 1975*; reviewed in *Chang, 2017*; *De Castro, 2009*; *Kumar and Prabhakar, 2012*; *Ortega-Sáenz and López-Barneo, 2020*). These acute reflexes are essential for optimizing tissue oxygenation of vital organs, including the brain, heart, and kidneys. However, in chronic conditions such as sleep-disorder breathing, hypertension, chronic heart failure, airway constriction, and metabolic syndrome, the CB becomes hyperactive, leading to exaggerated responses to hypoxia and sympathetic overactivity. Under these pathological conditions, suppressing CB activity improves causal symptoms such as hypertension (*Abdala et al., 2012*; *Del Rio et al., 2016*; *Fletcher et al., 1992*; *Narkiewicz et al., 2016*), cardiac arrhythmias (*Del Rio et al., 2013*; *Marcus et al., 2014*), and insulin resistance (*Ribeiro et al., 2013*; *Sacramento et al., 2017*). Thus, understanding the fundamental mechanisms of oxygen sensing in the CB is of considerable scientific and medical importance.

In a long-standing model, acute oxygen sensing in the CB is proposed to be mediated by the mitochondrial electron transport chain (ETC) in glomus cells (*Chang, 2017*; *Holmes et al., 2018*; *Ortega-Sáenz and López-Barneo, 2020*). In his discovery of the CB as a chemoreceptor in the 1920s, Corneille Heymans utilized cyanide to inhibit ETC complex IV (CIV) and mimic the effect of hypoxia (*Heymans, 1963*). More recently, genetic approaches in mice found that knockout of two ETC subunit genes attenuates CB sensory activity: the mitochondrial respiratory chain complex I (CI) core subunit *Ndufs2* and the HIF2A-regulated mitochondrial respiratory chain CIV subunit *Cox4i2* (*Fernández-Agüera et al., 2015*; *Moreno-Domínguez et al., 2020*). *Ndufs2* is ubiquitously expressed and essential for CI activity, whereas *Cox4i2* expression is limited to several tissues, including the lung, placenta, heart, tongue, breast, and adipose tissue (*Shen et al., 2012*; *Uhlén et al., 2015*; *Figure 1—figure supplement 1A–D*). These results are accompanied by the observations that the ETC of glomus cells is more sensitive to hypoxia compared to other cell types (*Buckler and Turner, 2013*; *Duchen and Biscoe, 1992a*; *Duchen and Biscoe, 1992b*; *Forster, 1968*; *Jobsis, 1968*) and the proposal that an unusual CIV contributes to oxygen sensitivity of the CB (*Mills and Jöbsis, 1970*; *Mills and Jöbsis, 1972*). However, given the breadth of *Cox4i2* expression, it remains unclear whether HIF2a regulation of *Cox4i2* is solely responsible for the exquisite sensitivity of glomus cell mitochondria to hypoxia compared to other tissues.

Here, we identify HIGD1C as a novel mitochondrial protein associated with ETC CIV that is almost exclusively expressed in the CB. HIGD1C is critical in mediating CB oxygen sensing and metabolic responses to hypoxia in mice. In heterologous cell culture, we demonstrate that HIGD1C regulates the activity and conformation of CIV, and when co-expressed with COX4I2, HIGD1C enhances the oxygen sensitivity of CIV to hypoxia. We propose that HIGD1C and COX4I2 comprise key components bestowing CB glomus cell mitochondria with their extreme oxygen sensitivity.

## Results
### HIGD1C is a novel mitochondrial protein expressed in CB glomus cells

CB sensory activity correlates with changes in ETC activity (*Chang, 2017*; *Holmes et al., 2018*; *Ortega-Sáenz and López-Barneo, 2020*). Therefore, we sought to identify proteins that are specifically expressed in mitochondria in the CB and are highly sensitive to changes in oxygen availability. Using whole-genome expression data from RNAseq, we looked for genes encoding putative mitochondrial proteins that are overexpressed in the adult mouse CB compared to the adrenal medulla, a similar but less oxygen-sensitive tissue (*Chang et al., 2015*). We found that three such genes, *Higd1c, Cox4i2*, and *Ndufa4l2*, were expressed at higher levels in the mouse CB (*Figure 1A*). The expression of *Cox4i2* and *Ndufa4l2* in the CB is found in glomus cells and regulated by *Hif2a*, a hypoxia-inducible transcription factor critical for CB development and function (*Macias et al., 2018*; *Moreno-Domínguez et al., 2020*; *Zhou et al., 2016*). We focused this study on *Higd1c* because it was the most differentially expressed of these genes and one of the top 10 most upregulated genes genome-wide in the mouse CB (*Chang et al., 2015*). RT-qPCR analysis confirmed the enrichment of *Higd1c* as well as *Ndufa4l2* and *Cox4i2* mRNAs in the human CB (*Figure 1B*).

*Higd1c* is a novel member of the HIG1 hypoxia-inducible domain gene family that also includes *Higd1a, Higd1b*, and *Higd2a*. *Higd1a* and *Higd2a*, the mammalian orthologs of the yeast respiratory

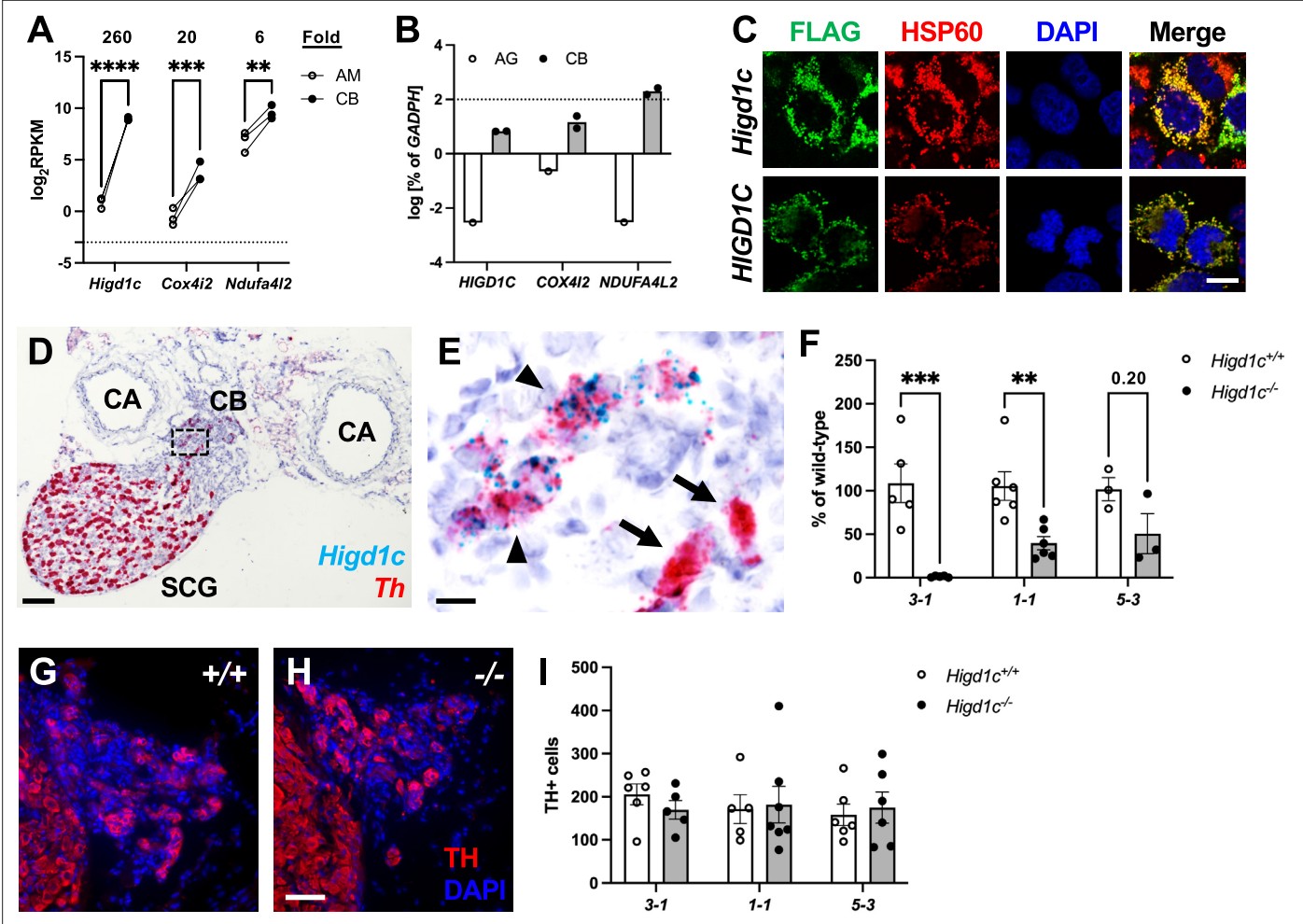

**Figure 1.** *Higd1c* expression in carotid body glomus cells is reduced in *Higd1c* CRISPR mutants. (**A**) Expression of genes encoding atypical mitochondrial electron transport chain (ETC) subunits in mouse carotid body (CB) versus adrenal medulla (AM) (***Chang et al., 2015***). RPKM, reads per kilobase of transcript, per million reads mapped. n = 3 cohorts of 10 animals each. Data as mean ± SEM. **p<0.01, ***p<0.001, ****p<0.0001 by two-way ANOVA with Sidak correction. (**B**) Expression of atypical ETC proteins in human CB and adrenal gland (AG). AG, one RNA sample of adrenal glands pooled from 62 individuals. CB, two RNA samples of CBs from two adults. Dotted line, 100% of *GAPDH* expression. Data as mean. (**C**) FLAG-tagged mouse and human HIGD1C (green) overexpressed in HEK293T cells co-localized with the mitochondrial marker HSP60 (red) by immunostaining. DAPI, nuclear marker. Scale bar, 10 µm. (**D, E**) BaseScope in situ hybridization of a wild-type C57BL/6J carotid bifurcation. (**E**) Boxed region from (**D**). SCG, superior cervical ganglion; CA, carotid arteries. Arrowheads, glomus cells. Arrows, SCG neurons. Scale bar, 100 µm (**D**), 10 µm (**E**). (**F**) Expression of *Higd1c* mRNA is reduced in CBs from *Higd1c* mutants measured by RT-qPCR. n = 3–6 samples. Each sample was prepared from 4 CBs/2 animals. Data as mean ± SEM. **p<0.01 by two-way ANOVA with Sidak correction. (**G, H**) Immunostaining of CB glomus cells. TH, tyrosine hydroxylase. DAPI, nuclear marker. Scale bar, 50 µm. (**I**) Quantitation of TH+ cells found no significant differences between CBs from *Higd1c^{+/+}* and *Higd1c^{-/-}* animals of each allele or between alleles by two-way ANOVA with Sidak correction (p>0.05). n = 5–7 CBs from 3-7 animals. Data as mean ± SEM.

The online version of this article includes the following source data and figure supplement(s) for figure 1:

**Source data 1.** Source data for ***Figure 1A, B, F, and I***.

**Figure supplement 1.** Tissue expression of genes encoding select mitochondrial proteins implicated in carotid body oxygen sensing.

**Figure supplement 1—source data 1.** Source data for ***Figure 1—figure supplement 1***.

**Figure supplement 2.** Generation of *Higd1c* CRISPR/Cas9 mutants.

**Figure supplement 2—source data 1.** Source gels for ***Figure 1—figure supplement 2C***.

**Figure supplement 3.** Expression of *Higd1c* in mouse tissues.

**Figure supplement 3—source data 1.** Source data for ***Figure 1—figure supplement 3***.

**Figure supplement 4.** *Higd1c* is expressed in carotid body glomus cells and kidney proximal tubules.

**Figure supplement 4—source data 1.** Source data for ***Figure 1—figure supplement 4K***.

*Figure 1 continued on next page*

*Figure 1 continued*

**Figure supplement 5.** Cellular and tissue expression of *Higd1c* in rodents and human.

**Figure supplement 5—source data 1.** Source data for *Figure 1—figure supplement 5*.

supercomplex factors 1 and 2 (*Rcf1* and *Rcf2*), encode mitochondrial proteins that promote the biogenesis of ETC complexes and their assembly into supercomplexes (*Timón-Gómez et al., 2020a*). To determine the subcellular localization of HIGD1C, we overexpressed FLAG-tagged HIGD1C in HEK293T cells and observed that it co-localizes with the mitochondrial marker HSP60, suggesting that HIGD1C is targeted to mitochondria like HIGD1A and HIGD2A (*Figure 1C*).

Compared to mitochondrial ETC genes previously implicated in CB oxygen sensing (*Ndufs2* and *Cox4i2*), mRNA transcripts for *Higd1c* are minimally detected across mouse and human tissues, with the exception of the mouse kidney (*An et al., 2011*; *Shen et al., 2012*; *Uhlén et al., 2015*; *Figure 1—figure supplement 1A–D*). We found that *Higd1c* is expressed at 30–600,000-fold higher levels in the CB than in other mouse tissues (*Figure 1—figure supplement 2A–D*, *Figure 1—figure supplement 3A–D*). Within the CB, glomus cells sense hypoxia to stimulate afferent nerves to increase ventilation *Kumar and Prabhakar, 2012*. In situ hybridization showed that *Higd1c* mRNA was localized in the same cells as mRNA for *Th*, a marker of glomus cells (*Figure 1D and E*, *Figure 1—figure supplement 4A–D*), validating single-cell RNAseq findings (*Zhou et al., 2016*; *Figure 1—figure supplement 5A*). In the rat, *Higd1c* was also expressed at higher levels in the CB compared to the neonatal and adult adrenal medulla and thoracic spinal cord, which contains a novel central oxygen sensor (*Barioni et al., 2022*; *Figure 1—figure supplement 5B*). Additionally, we confirmed the expression of *Higd1c* mRNA in mouse kidney proximal tubules as previously reported (*Suganthan et al., 2014*; *Figure 1—figure supplement 4F–I*). These results indicate that *Higd1c* is enriched in a population of cells in the CB essential for oxygen sensing.

To determine whether HIGD1C plays a role in CB oxygen sensing, we generated mutants in *Higd1c* by CRISPR/Cas9 in C57BL/6J mice. We isolated F0 mice that carried large deletions that span upstream sequences through the first coding exon and small indels in the first coding exon (*Figure 1—figure supplement 2A–C*). We characterized three alleles representing large deletions (*3-1*) and early frameshift mutations in both frames downstream of the start codon (*1-1* and *5-3*). The *3-1* allele was predicted to either produce no protein or a truncated protein missing the N-terminus while the *1-1* and *5-3* alleles were expected to make truncated proteins with early amino acid changes (*Figure 1—figure supplement 2D*). For all three alleles, heterozygous *Higd1c* mutant mice were fertile, and homozygous mutants were viable and not underrepresented in the progeny (*Table 1*). The large deletion allele *3-1* was used as a negative control in characterizing *Higd1c* expression by RT-qPCR and in situ hybridization because our primers and probes targeted a region that was deleted in this allele (*Figure 1F*, *Figure 1—figure supplement 3A–D*, *Figure 1—figure supplement 4E and J*). In *1-1* and *5-3* alleles, *Higd1c* mRNA levels were reduced by 40–90% in CBs and kidneys from *Higd1c^{-/-}* mutants compared to *Higd1c^{+/+}* animals (*Figure 1F*, *Figure 1—figure supplement 4K*). While we infer that all three alleles alter HIGD1C protein sequence, *Higd1c 1-1* and *5-3* alleles also have reduced levels of HIGD1C, perhaps due to nonsense-mediated mRNA decay.

## HIGD1C mediates CB sensory and metabolic responses to hypoxia

To assess whether HIGD1C plays a role in CB oxygen sensing at the whole animal level, we performed whole-body plethysmography on awake, unanesthetized mice. A decrease in arterial blood oxygen stimulates the CB to signal the brainstem to increase ventilation within seconds (*Chang, 2017*; *Kumar and Prabhakar, 2012*; *Ortega-Sáenz and López-Barneo, 2020*). *Higd1c^{-/-}* mutants of all three alleles had normal ventilation in normoxia but were similarly defective in the hypoxic ventilatory response (*Figure 2A–D*, *Figure 2—figure supplement 1A–M*). The defects observed in these *Higd1c* alleles were at least as severe as ablation or denervation of the CBs in rodents (*Del Rio et al., 2013*; *Soliz et al., 2005*). By contrast,

**Table 1.** Genotype distribution of progeny from *Higd1c^{+/-}* × *Higd1c^{+/-}* crosses.

| *Higd1c* allele | N | Frequency of genotype | | | Chi-square p-value |
|---|---|---|---|---|---|
| | | +/+ | +/- | -/- | |
| *3-1* | 125 | 0.22 | 0.48 | 0.30 | 0.344 |
| *1-1* | 233 | 0.32 | 0.43 | 0.25 | 0.028 |
| *5-3* | 78 | 0.24 | 0.53 | 0.23 | 0.891 |

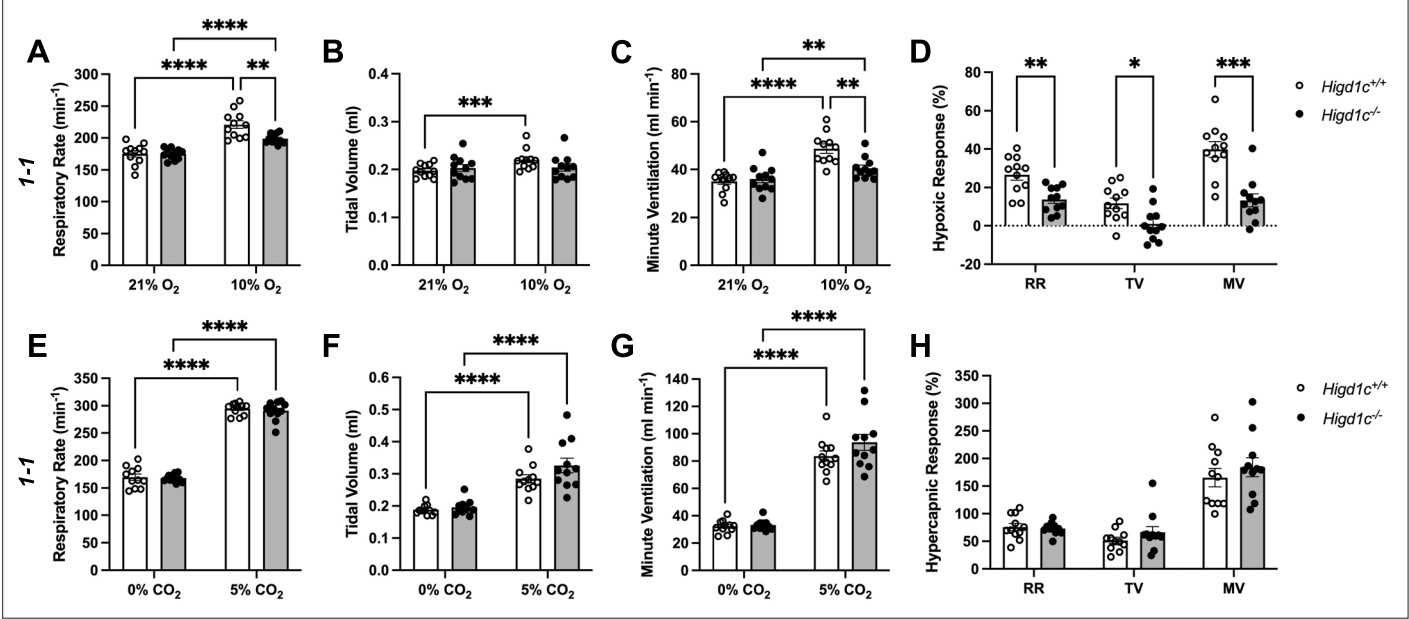

**Figure 2.** Ventilatory responses of *Higd1c* mutants to hypoxia and hypercapnia. (**A–H**) Respiratory rate (RR), tidal volume (TV), and minute ventilation (MV) (minute ventilation = respiratory rate × tidal volume) by whole-body plethysmography of unrestrained, unanesthetized *Higd1c 1-1⁺/⁺* and *Higd1c 1-1⁻/⁻* animals exposed to hypoxia (**A–D**) or hypercapnia (**E–H**). (**D**) Hypoxic response as the percentage change in hypoxia (10% $O_2$) versus control (21% $O_2$). (**H**) Hypercapnic response as the percentage change in hypercapnia (5% $CO_2$) versus control (0% $CO_2$). n = 11 (+/+), 11 (-/-) animals. Data as mean ± SEM. *p<0.05, **p<0.01, ***p<0.001, ****p<0.0001 by two-way repeated-measures ANOVA with Sidak correction (**A–C, E-G**) or unpaired *t*-tests (**D, H**) with Holm–Sidak correction. Ventilatory parameters of *Higd1c⁺/⁺* and *Higd1c⁻/⁻* animals in normal air conditions (21% $O_2$ or 0% $CO_2$) were not significantly different (p>0.05). For the hypoxic response (**D**), Cohen's *d* = 1.24 (RR), 1.20 (TV), 2.13 (MV).

The online version of this article includes the following source data and figure supplement(s) for figure 2:

**Figure supplement 1.** Ventilatory responses of *Higd1c* mutants to hypoxia.

**Figure supplement 2.** Ventilatory responses of *Higd1c* mutants to hypercapnia.

**Source data 1.** Source data for *Figure 2*.

**Figure supplement 1—source data 1.** Source data for *Figure 1—figure supplement 1*.

**Figure supplement 2—source data 1.** Source data for *Figure 1—figure supplement 2*.

*Higd1c⁻/⁻* mice maintained robust ventilatory responses to hypercapnia comparable to *Higd1c⁺/⁺* animals (*Figure 2E–H*, *Figure 2—figure supplement 2A–J*). These results suggest that *Higd1c* specifically regulates ventilatory responses to hypoxia.

Because *Higd1c* was expressed at low levels in the petrosal ganglion and brainstem downstream of the CB in the neuronal circuit (*Figure 1—figure supplement 3A and B*), the reduction in hypoxic ventilatory response in *Higd1c⁻/⁻* mice was most likely due to loss of HIGD1C activity in the CB. When we examined the CB, the number of TH-positive glomus cells from *Higd1c⁺/⁺* and *Higd1c⁻/⁻* animals was not significantly different for all three alleles (*Figure 1G–I*), and there were no gross morphological abnormalities in mutant CBs. Thus, it is unlikely that the hypoxic ventilatory response defect observed in *Higd1c⁻/⁻* mutants is due to a loss of glomus cells.

Next, we measured the integrated sensory output from the CB at the level of the carotid sinus nerve (CSN), the nerve that transduces signals from the CB to the brainstem (*Kumar and Prabhakar, 2012*). Baseline CSN activity was similar between *Higd1c 1-1⁺/⁺* and *Higd1c 1-1⁻/⁻* tissues (*Figure 3A–C*). As oxygen levels were decreased, CSN activity increased in a dose-dependent manner in *Higd1c 1-1⁺/⁺* tissue (*Figure 3A and D*). However, while this response to hypoxia was attenuated in CSNs from *Higd1c 1-1⁻/⁻* mutants (*Figure 3B and D*), the response to high $CO_2$/$H⁺$ was unaffected (*Figure 3A–D*). Thus, we conclude that HIGD1C specifically mediates oxygen sensing at the level of the whole CB organ.

Because *Higd1c* was expressed in glomus cells (*Figure 1D and E*), we evaluated whether *Higd1c* mutants were also defective in the sensory responses of these oxygen-sensitive cells. Glomus cells

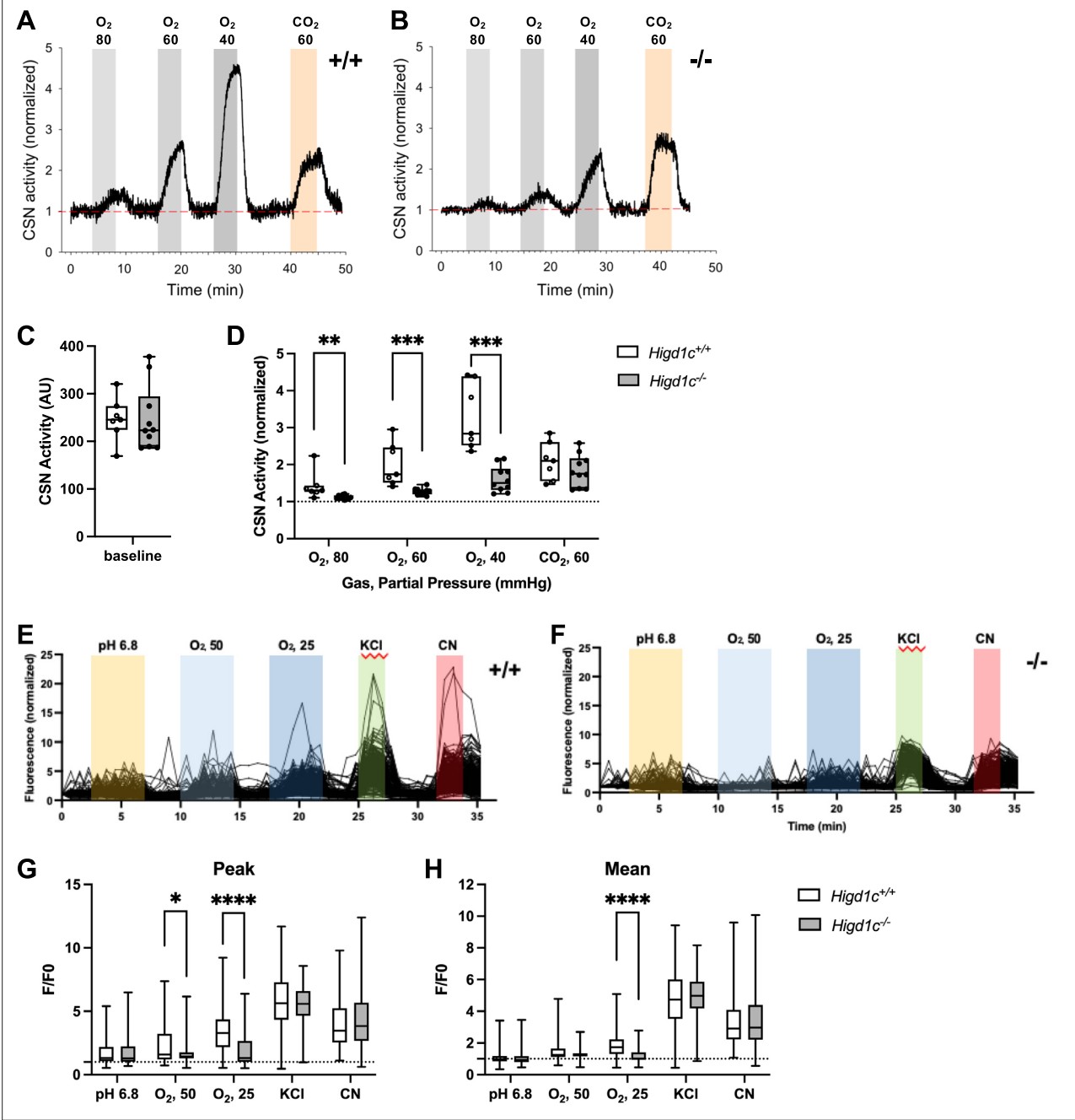

**Figure 3.** *Higd1c* mutants have defects in carotid body sensory responses to hypoxia. (**A, B**) Representative traces of carotid sinus nerve (CSN) activity from *Higd1c 1-1+/+* and *Higd1c 1-1-/-* tissue preparations exposed to hypoxia (PO₂ ~ 80, 60, and 40 mmHg) or hypercapnia (PCO₂ ~ 60 mmHg) normalized to activity at t = 0. (**C, D**) CSN activity in *Higd1c 1-1+/+* and *Higd1c 1-1-/-* tissues at baseline (**C**) and in hypoxia and hypercapnia (**D**). Activity in hypoxia and hypercapnia normalized to baseline (**D**). n = 7/5 (+/+), 10/6 (-/-) preparations/animals. AU, arbitrary units. Data as box plots showing median and interquartile interval. **p<0.01, ***o<0.001 by Mann–Whitney *U*-test with Holm–Sidak correction. (**E, F**) Individual traces of GCaMP fluorescence of glomus cells from *Higd1c 1-1+/+* and *Higd1c 1-1-/-* animals, in response to low pH (6.8), hypoxia (PO₂ ~ 50 and 25 mmHg), high KCl (40 mM), and cyanide (CN, 1 mM). Data normalized to fluorescence at t = 0 s. Z stacks were collected every 45 s. (**G, H**) Peak (**G**) and mean (**H**) GCaMP calcium responses (F/F0) of glomus cells from *Higd1c 1-1+/+* and *Higd1c 1-1-/-* animals. n = 296/4/3 (+/+), 201/4/3 (-/-) for pH 6.8, 312/5/4 (+/+), 214/5/4 (-/-) for all other stimuli. n as glomus cells/CBs/animals. Data as box plots showing median and interquartile interval. *p<0.05, ****p<0.0001 by Mann–Whitney *U*-test with Holm–Sidak correction.

The online version of this article includes the following source data and figure supplement(s) for figure 3:

**Source data 1.** Source data for *Figure 3C, D, G and H*.

*Figure 3 continued on next page*

*Figure 3 continued*

**Figure supplement 1.** Calcium responses of *Higd1c 1-1* glomus cells to hypoxia.

**Figure supplement 1—source data 1.** Source data for *Figure 3—figure supplement 1*.

exhibit acute calcium transients in response to stimuli, which can be visualized by the genetically encoded calcium indicator GCaMP3 (*Chang et al., 2015*). We found that glomus cells from *Higd1c 1-1*$^{-/-}$ mutants mounted a weaker calcium response to hypoxia than those from *Higd1c 1-1*$^{+/+}$ animals, with fewer glomus cells responding strongly to both levels of hypoxia (*Figure 3E–H*, *Figure 3—figure supplement 1A–E*). Low pH and high KCl modulate the activity of ion channels on the plasma membrane of glomus cells thought to act downstream of mitochondria in CB oxygen sensing (*Buckler, 2015*; *Lu et al., 2013*). In contrast to hypoxia, glomus cells from *Higd1c 1-1*$^{-/-}$ mutants were not significantly different from *Higd1c 1-1*$^{+/+}$ animals in their calcium response to low pH or high KCl compared to glomus cells from *Higd1c 1-1*$^{-/-}$ animals (*Figure 3E–H*). Calcium responses to cyanide, a potent ETC CIV inhibitor, were also similar between *Higd1c 1-1*$^{+/+}$ and *Higd1c 1-1*$^{-/-}$ glomus cells (*Figure 3E–H*), suggesting that strong ETC inhibition is still able to trigger sensory responses in glomus cells mutated in *Higd1c* like other mutants with defects in CB oxygen sensing (*Peng et al., 2020*; *Peng et al., 2019*). Together, these results show that HIGD1C contributes specifically to glomus cell responses to hypoxia.

To determine whether HIGD1C modulates oxygen sensitivity of mitochondria in glomus cells, we used rhodamine 123 (Rh123) to image the inner mitochondrial membrane (IMM) potential generated by ETC activity. Hypoxia inhibits ETC activity, leading to a decrease in IMM potential and an increase in Rh123 fluorescence (*Perry et al., 2011*). *Higd1c 1-1*$^{-/-}$ glomus cells had an attenuated response to hypoxia and a higher percentage of cells that responded poorly to FCCP, a potent uncoupler of oxidative phosphorylation that depolarizes the IMM (*Figure 4A–C*, *Figure 4—figure supplement 1A and B*). The weaker response of *Higd1c 1-1*$^{-/-}$ glomus cells to FCCP was evident even when FCCP was presented as the first stimulus (*Figure 4—figure supplement 1C*), suggesting that the IMM is less polarized at baseline in mutant glomus cells. This idea is also supported by the observation that in glomus cells that had a strong FCCP response > 0.2, Rh123 fluorescence was greater in *Higd1c 1-1*$^{-/-}$ glomus cells in normoxia ($PO_2$ = 100 mmHg) (*Figure 4D*). Nevertheless, the increase in fluorescence in hypoxia ($PO_2$ < 80 mmHg) was smaller than in *Higd1c 1-1*$^{+/+}$ glomus cells (*Figure 4E and F*). This pattern of weaker ETC activity in normoxia that cannot be suppressed further in hypoxia seen in *Higd1c 1-1*$^{-/-}$ glomus cells resembles that of acute CII inhibition on CB sensory activity (*Swiderska et al., 2021*). In *Higd1c 1-1*$^{+/+}$ CBs, we found that vascular cells, which are less oxygen-sensitive than glomus cells, had a left-shifted IMM potential response to hypoxia (*Figure 4G–I*), indicating that our hypoxic stimulus was in an appropriate range to detect the enhanced oxygen sensitivity of the CB over other cell types. These results demonstrate that HIGD1C enhances ETC inhibition by hypoxia in glomus cells, a response linked to CB sensory activity (*Chang, 2017*; *Holmes et al., 2018*; *Ortega-Sáenz and López-Barneo, 2020*).

## HIGD1C associates with and regulates ETC complex IV activity

To assess the role of HIGD1C in ETC function, we overexpressed FLAG-tagged human or mouse HIGD1C in HEK293T cells and performed biochemical and metabolic studies of mitochondria. In wild-type HEK293T cells, *HIGD1C* mRNA was expressed at very low levels ($3 \times 10^{-6}$ the level of *GAPDH* by RT-qPCR). Overexpressed FLAG-tagged HIGD1C associated with ETC CIV and cytochrome *c* (*Figure 5—figure supplement 1A–C*). Notably, HIGD1C overexpression severely reduced the abundance of ETC supercomplexes, and supercomplexes that did assemble contained only traces of *in-gel* CIV activity (*Figure 5—figure supplement 1D and E*). These defects in supercomplex formation correlated with a decrease in CIV enzymatic activity (*Figure 5—figure supplement 1F and G*) and oxygen consumption rate (OCR) (*Figure 5—figure supplement 1H and I*).

Because HIGD1C is most similar to HIGD1A and HIGD2A, we overexpressed HIGD1C in *HIGD1A*-KO and *HIGD2A*-KO HEK293T cell lines to determine whether it could rescue ETC defects of these KO cell lines (*Timón-Gómez et al., 2020b*). As in wild-type cells, HIGD1C associated with CIV and cytochrome *c* in *HIGD1A*-KO and *HIGD2A*-KO mutant cells (*Figure 5A and B*, *Figure 5—figure supplement 2A–D*, *Figure 5—figure supplement 3A–C*). Unlike HIGD1A, HIGD1C did not associate with CIII and could not rescue defects in the assembly of supercomplexes in either *HIGD1A*-KO or *HIGD2A*-KO cells (*Figure 5C*, *Figure 5—figure supplement 2E*, *Figure 5—figure supplement 3D*; *Timón-Gómez*

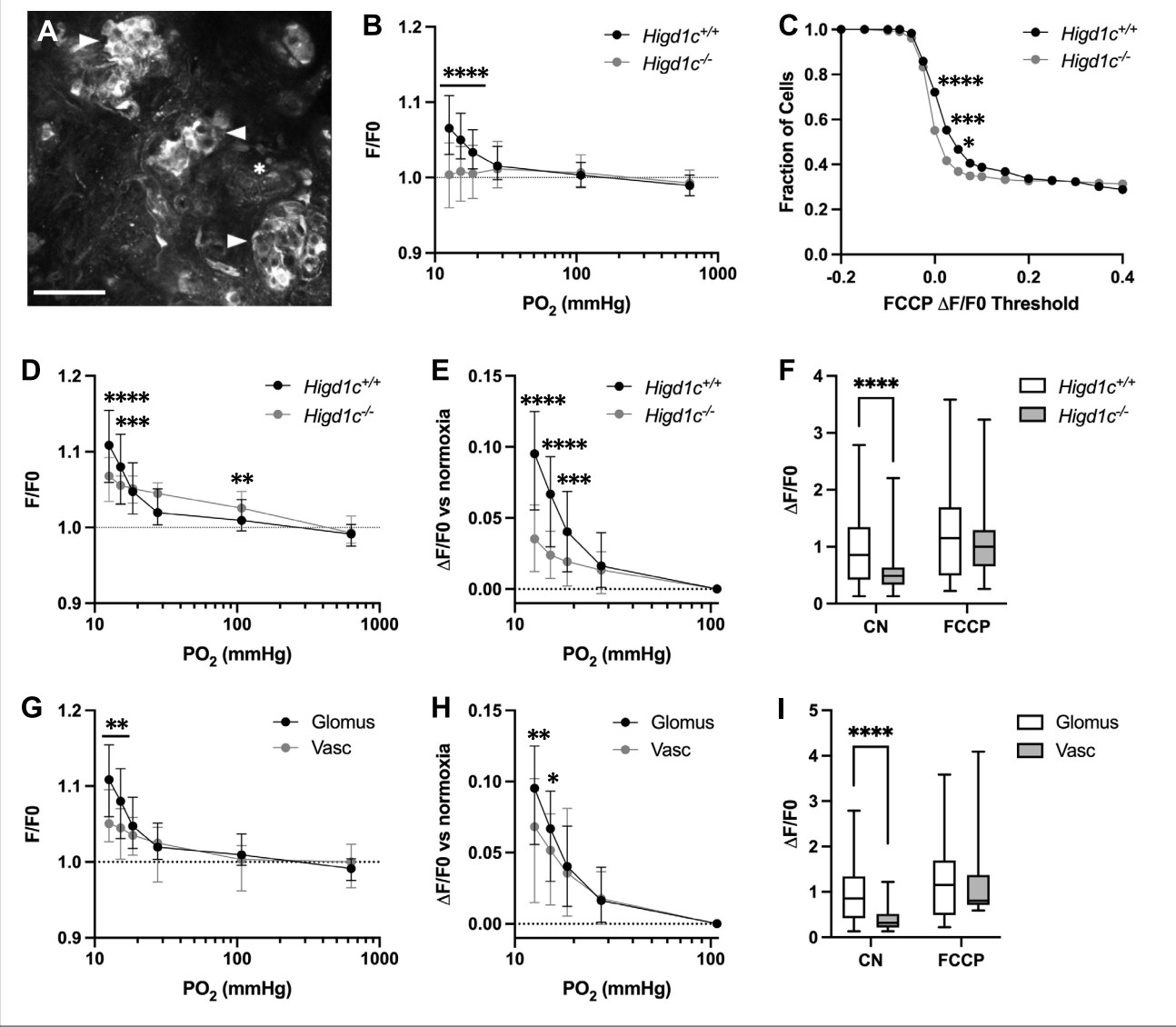

**Figure 4.** HIGD1C regulates the hypoxic response of the electron transport chain in carotid body glomus cells. (**A**) Fluorescence of a whole-mount carotid body (CB) loaded with rhodamine 123 (Rh123), a dye sensitive to changes in mitochondrial inner membrane potential, under quenching conditions. Arrowheads, glomus cell clusters; asterisk, vasculature. Scale bar, 50 µm. Rh123 fluorescence of glomus cells in *Higd1c 1-1*[+/+] and *Higd1c 1-1*[-/-] CBs measured in response to hypoxia (PO$_2$ < 80 mmHg), cyanide (1 mM), and FCCP (2 µM). (**B**) Rh123 response to hypoxia for all glomus cells quantified. Dashed line, fluorescence at the start of stimulus. n = 291/3/3 (+/+), 312/3/3 (-/-) glomus cells/CBs/animals. Data presented as the median and interquartile interval. ****p<0.0001 by Mann–Whitney *U*-test with Holm–Sidak correction. (**C**) Fraction of glomus cells that responded to FCCP at different ΔF/F. n = 291/3/3 (+/+), 312/3/3 (-/-) glomus cells/CBs/animals. *p<0.01, ***p<0.001, ****p<0.0001 by *Z*-test of proportions. (**D–F**) Rh123 response to hypoxia for glomus cells with FFCP responses of ΔF/*F* > 0.2. Dashed line, fluorescence at the start of stimulus. n = 98/3/3 (+/+), 102/3/3 (-/-) glomus cells/CBs/animals. Data presented as the median and interquartile interval or box plots. **p<0.01, ***p<0.001, ****p<0.0001 by Mann–Whitney *U*-test with Holm–Sidak correction. (**G–I**) Rh123 fluorescence of vascular cells in the CB compared to glomus cells with FCCP responses of ΔF/*F* > 0.2. n = 98/3/3 (+/+) glomus cells/CBs/animals, 49/3/3 (+/+) vascular cells/CBs/animals. Data presented as the median and interquartile interval or box plots. *p<0.05, **p<0.01, ****p<0.0001 by Mann–Whitney *U*-test with Holm–Sidak correction.

The online version of this article includes the following source data, source code, and figure supplement(s) for figure 4:

**Source code 1.** Source code for R analysis for *Figure 4*.

**Figure supplement 1.** Metabolic responses of *Higd1c 1-1* glomus cells to hypoxia.

**Source data 1.** Source data for *Figure 4B–I*.

**Figure supplement 1—source data 1.** Source data for *Figure 4—figure supplement 1*.

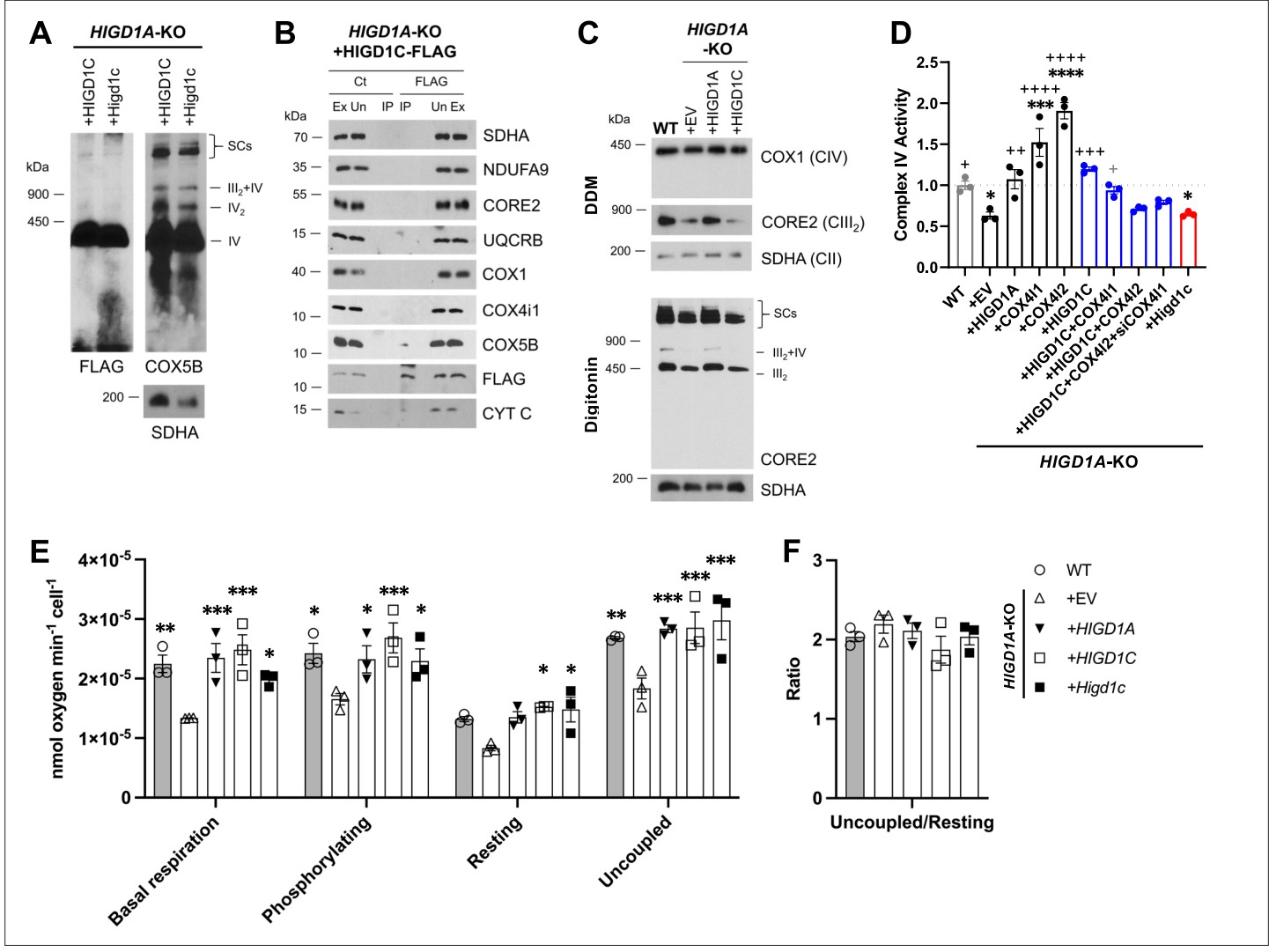

**Figure 5.** HIGD1C is a mitochondrial protein that associates with the electron transport chain complex IV and regulates cellular respiration. *HIGD1A*-KO HEK293T cells overexpressing FLAG-tagged human or mouse HIGD1C and/or COX4 isoforms. EV, empty vector; HIGD1C, human HIGD1C; Higd1c, mouse HIGD1C. (**A**) BN-PAGE and immunoblots using antibodies for FLAG and the complex IV subunit COX5B. SDHA is used as a loading control. (**B**) Co-immunoprecipitation using a FLAG antibody followed by SDS-PAGE and immunoblot using antibodies for FLAG and subunits of complex I (NDUFA9), complex II (SDHA), complex III (CORE2, UQCRB), complex IV (COX1, COX4I1, COX5B), and cytochrome *c*. (**C**) Electron transport chain (ETC) complexes and supercomplexes extracted with DDM and digitonin, respectively, detected by BN-PAGE and immunoblotting. (**A–C**) All gels and blots were repeated three times. (**D**) Complex IV enzymatic activity assay. n = 3. *p<0.05, ***p<0.001, ****p<0.0001 vs. WT by one-way ANOVA with Dunnett's test. +p<0.05, ++p<0.01, +++p<0.001, ++++p<0.0001 vs. *HIGD1A*-KO+EV by one-way ANOVA with Dunnett's test. Gray symbol indicates p=0.06. (**E, F**) Polarographic assessment in digitonin-permeabilized cells of KCN-sensitive oxygen consumption driven by succinate and glycerol-3-phosphate, in the presence or absence of ADP (basal respiration and phosphorylating), oligomycin (resting), and the uncoupler CCCP (uncoupled). Respiratory control ratio (**F**) of measurements performed in (**E**). n = 3. Data as mean ± SEM. *p<0.05, **p<0.01, ***p<0.001 vs. *HIGD1A*-KO+EV by two-way ANOVA with Dunnett's test.

The online version of this article includes the following source data and figure supplement(s) for figure 5:

**Figure supplement 1.** HIGD1C overexpression in wild-type HEK293T cells inhibits complex IV (CIV) activity.

**Figure supplement 2.** HIGD1C regulates complex IV (CIV) activity in *HIGD1A*-KO HEK293T cells.

**Figure supplement 3.** HIGD1C overexpression in *HIGD2A*-KO HEK293T cells does not rescue electron transport chain (ETC) defects.

**Source data 1.** Source blots for *Figure 5A–C*.

**Source data 2.** Source data for *Figure 5D–F*.

**Figure supplement 1—source data 1.** Source gels and blots for *Figure 5—figure supplement 1A–E*.

**Figure supplement 1—source data 2.** Source data for *Figure 5—figure supplement 1F–I*.

*Figure 5 continued on next page*

*Figure 5 continued*

**Figure supplement 2—source data 1.** Source blots for *Figure 5—figure supplement 2A–E*.

**Figure supplement 2—source data 2.** Source data for *Figure 5—figure supplement 2F*.

**Figure supplement 3—source data 1.** Source blots for *Figure 5—figure supplement 3A–D*.

**Figure supplement 3—source data 2.** Source data for *Figure 5—figure supplement 3E and F*.

---

*et al., 2020b*). Strikingly, however, HIGD1C overexpression restored CIV activity in *HIGD1A*-KO, but not *HIGD2A*-KO cells (*Figure 5D*, *Figure 5—figure supplement 2F*, *Figure 5—figure supplement 3E and F*). This could be due to non-overlapping activities of HIGD1C and HIGD2A and/or more severe defects in CIV assembly in *HIGD2A*-KO cells (*Figure 5—figure supplement 3D*). Instead of acting as a CIV assembly factor as HIGD2A, HIGD1C could play a regulatory role in modulating CIV activity like HIGD1A (*Timón-Gómez et al., 2020b*). Supporting this idea, we observed that cellular respiration at the overall ETC level was also restored by HIGD1C overexpression in *HIGD1A*-KO cells (*Figure 5E and F*). Overexpression of mouse HIGD1C induced a weaker rescue of CIV activity than human HIGD1C, likely due to disruption of species-specific associations between ETC subunits (*Figure 5D*, *Figure 5—figure supplement 2F*). Nonetheless, mouse HIGD1C fully rescued mitochondrial oxygen consumption due to the spare respiratory capacity of the ETC (*Figure 5E and F*). These rescue experiments show that HIGD1C is not involved in ETC complex or supercomplex biogenesis, but similarly to HIGD1A, it can interact with CIV to regulate its activity.

HIGD1C could modulate CIV activity by (1) mediating the formation of an electron-transfer bridge between ETC CIII and IV and/or (2) changing the structure around the active center of the enzyme. The former possibility is unlikely because overexpression of HIGD1C in *HIGD1A*-KO cells did not increase the levels of cytochrome *c* present in ETC supercomplexes compared to control cell lines (*Figure 5—figure supplement 2E*). The CIV active center that reduces oxygen to water in the terminal step of the ETC is located in subunit 1 (COX1) and formed by a binuclear heme-copper center (heme $a_3$-Cu$_B$) (*Timón-Gómez et al., 2018*). To analyze the environment around the CIV active center, we measured UV/Vis absorption spectra of total cytochromes extracted from purified mitochondria. The absence of HIGD1A produced a blue shift in the peak of heme *a+a3* absorbance from 603 nm to 599 nm (*Figure 6A*, *Figure 6—figure supplement 1A*) that is associated with changes around the CIV heme *a* centers (*Shapleigh et al., 1992*). Previous studies showed that adding excess recombinant HIGD1A to highly purified oxidized CIV increases CIV activity twofold and changes the conformation around the heme *a* center (*Hayashi et al., 2015*), suggesting that HIGD1A levels modulate CIV activity. Here, the spectral shift observed in the *HIGD1A*-KO cell line was completely restored by expressing HIGD1A (*Figure 6A*, *Figure 6—figure supplement 1A*). While the wavelength at the peak appeared to be restored by human HIGD1C, expression of human or mouse HIGD1C generated a broader peak, probably due to the existence of a mixed population of the enzyme, to partially restore the spectral shift (*Figure 6A–D*, *Figure 6—figure supplement 1A–C*). A blue shift in the wavelength at the peak was apparent in *HIGD1A*-KO cells overexpressing the mouse HIGD1C alone or human HIGD1C and COX4I2 together (*Figure 6A–D*, *Figure 6—figure supplement 1A–C*), suggesting that the atypical CIV subunit COX4I2 expressed in the CB (*Figure 1A and B*, *Figure 1—figure supplement 5A and C*) can modify the effect of HIGD1C overexpression. In addition, this spectral shift resembled the unusual absorbance spectrum of cytochromes found in the CB (*Streller et al., 2002*). These results suggest that interaction of HIGD1C with CIV can alter the active site and activity of CIV.

## HIGD1C and COX4I2 enhance the sensitivity of ETC complex IV to hypoxia

To determine whether HIGD1C can modify the sensitivity of ETC to hypoxia, we measured respiration of *HIGD1A*-KO cells overexpressing atypical CIV proteins found in glomus cells (*Zhou et al., 2016*; *Figure 1—figure supplement 5A*). Because CIV activity alone is sufficient to recapitulate the enhanced oxygen sensitivity of the intact ETC in glomus cells (*Buckler and Turner, 2013*), we used an artificial electron donor system to isolate cytochrome *c*-CIV activity and measured oxygen consumption. This approach allowed us to bypass CIII assembly defects of *HIGD1A*-KO cells that contribute to defects in total respiration and measure CIV activity derived from the cyanide-dependent component of total respiration (*Figure 5D and E*; *Timón-Gómez et al., 2020b*). Co-expression of HIGD1C and COX4I2

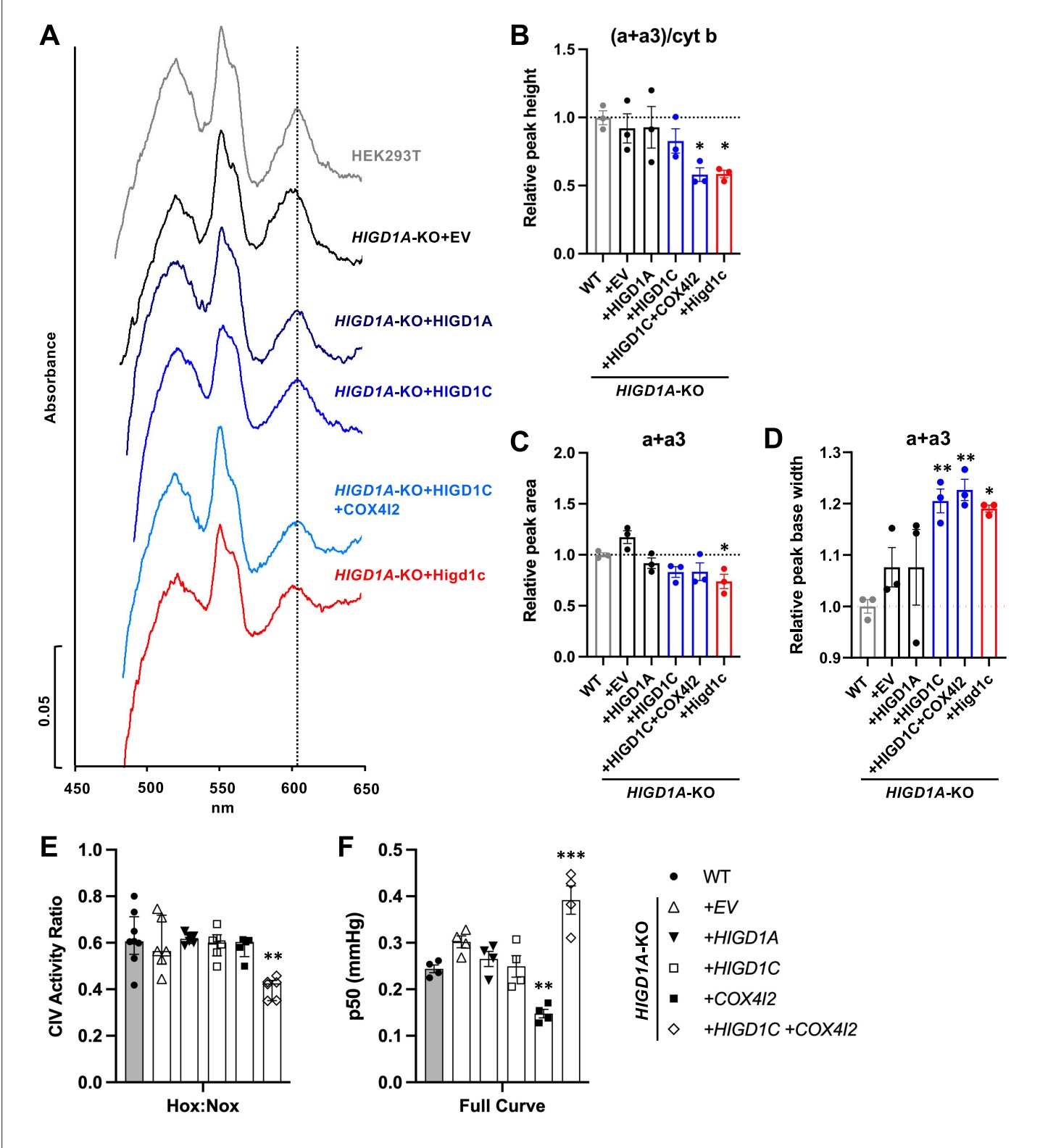

**Figure 6.** HIGD1C alters complex IV (CIV) conformation and increases CIV sensitivity to hypoxia. (**A**) Differential spectra (reduced minus oxidized) of total mitochondrial cytochromes measured by spectrophotometry. The absorbance of cytochromes extracted from purified mitochondria was measured from 450 to 650 nm. (**B**) Relative peak height of heme *a + a3* normalized by cytochrome *b* peak as the ratio of wild-type (WT). n = 3. Data as mean ± SEM. **p<0.01 vs. WT by one-way ANOVA with Dunnett's test. (**C, D**) Relative peak area (**C**) and peak base width (**D**) of *a + a₃* as the ratio of WT. n =

*Figure 6 continued on next page*

*Figure 6 continued*

3. Data as mean ± SEM. *$p<0.05$, **$p<0.01$ vs. WT by one-way ANOVA with Dunnett's test. (**E**) Ascorbate/TMPD-dependent oxygen consumption in normoxia (Nox, $PO_2$ ~ 150 mmHg) and hypoxia (Hox, $PO_2$ ~ 25 mmHg) by high-resolution respirometry. Ratio of hypoxic/normoxic oxygen consumption. n = 5–8. Data as the median and interquartile interval. **$p<0.01$ vs. WT by Kruskal–Wallis test with Dunn's test. (**F**) Mitochondrial oxygen affinity (p50$_{mito}$) values derived from full oxygen consumption curve in intact cells from normoxia ($PO_2$ ~ 150 mmHg) to anoxia ($PO_2$ = 0 mmHg) by high-resolution respirometry. n = 4. Data as mean ± SEM. **$p<0.01$, ***$p<0.001$ vs. WT by one-way ANOVA with Dunnett's test.

The online version of this article includes the following source data and figure supplement(s) for figure 6:

**Figure supplement 1.** HIGD1C regulates complex IV (CIV) conformation and oxygen sensitivity in *HIGD1A*-KO HEK293T cells.

**Source data 1.** Source data for *Figure 6B–F*.

**Figure supplement 1—source data 1.** Source data for *Figure 6—figure supplement 1B–D*.

in *HIGD1A*-KO cells, which better models the ETC composition in the CB (*Figure 1—figure supplement 5A and C*), decreased CIV-dependent respiration in hypoxia more than wild-type and other cell lines, including one overexpressing HIGD1C alone (*Figure 6E*, *Figure 6—figure supplement 1D*). This condition also increased the oxygen pressure at half-maximal respiration (p50$_{mito}$), suggesting a reduction in oxygen affinity (*Figure 6F*, *Figure 6—figure supplement 1E*). In *HIGD1A*-KO cells, overexpression of COX4I2 alone decreased p50$_{mito}$, but additional expression of HIGD1C further modified the JO$_2$/Jmax curve to increase p50$_{mito}$ over wild-type (*Figure 6F*, *Figure 6—figure supplement 1E*). These results demonstrate that co-overexpression of HIGD1C and COX4I2, two atypical mitochondrial ETC proteins expressed in CB glomus cells that mediate oxygen sensing (*Moreno-Domínguez et al., 2020*), can confer hypersensitivity to hypoxia in HEK293T cells. Therefore, the COX4I2-containing CIV, and its regulation by HIGD1C, emerge as necessary and sufficient factors to promote oxygen sensing by CB glomus cells.

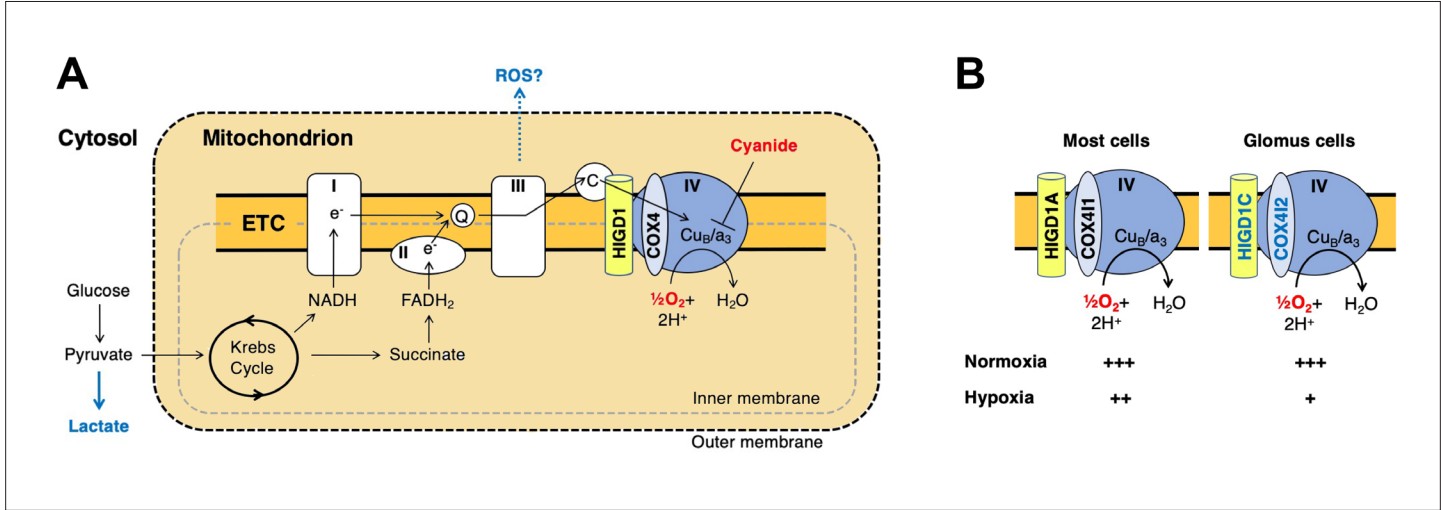

**Figure 7.** A model for oxygen sensing by mitochondria of carotid body glomus cells. (**A**) In this simplified scheme, NADH produced by the Krebs cycle transfers electrons to CI to initiate the electron transport chain (ETC). FADH$_2$ produced by succinate metabolism can also initiate the ETC by donating electrons to CII. In the terminal step of the ETC, complex IV (CIV) transfers electrons to oxygen. Cyanide inhibits the transfer of electrons to oxygen by binding to heme a$_3$ in CIV to mimic the effect of hypoxia on the ETC. Hypoxia and cyanide reduce flux through the ETC, increasing the production of reactive oxygen species (ROS) and lactate that are proposed to signal to downstream targets for neurotransmission in glomus cells (*Chang, 2017*; *Holmes et al., 2018*; *Ortega-Sáenz and López-Barneo, 2020*). HIGD1 and COX4 are ETC proteins that associate with CIV. Q, coenzyme Q; C, cytochrome *c*. (**B**) In most cells, CIV contains HIGD1A and COX4I1 proteins that form an early-assembly module during CIV biogenesis (*Timón-Gómez et al., 2020b*). Glomus cells express alternative isoforms of HIGD1A and COX4I2 called HIGD1C and COX4I2, respectively (*Figure 1—figure supplement 5A*). The combination of HIGD1C and COX4I2 increases the sensitivity of CIV to hypoxia at the level of oxygen consumption (relative activity levels denoted). Because mouse knockouts in *Higd1c* and *Cox4i2* are defective in carotid body oxygen sensing (*Figures 2–4 Moreno-Domínguez et al., 2020*) and HIGD1C and COX4I2 overexpression in HEK293T cells enhances oxygen sensitivity of the ETC to hypoxia (*Figure 6E and F*), we propose that these CIV-associated proteins are necessary and sufficient for oxygen sensing by carotid body glomus cells.

## Discussion

Previous studies found that mouse knockouts in specific CI (*Ndufs2*) and CIV (*Cox4i2*) subunits exhibit defects in CB sensory and metabolic responses to hypoxia (*Fernández-Agüera et al., 2015*; *Moreno-Domínguez et al., 2020*), phenocopying the effect of drugs that inhibit these ETC complexes. However, these subunits are expressed in multiple tissues in addition to the CB. Here, we identified HIGD1C as a novel mitochondrial CIV protein expressed almost exclusively in CB glomus cells that is essential for oxygen sensing by the CB (summarized in *Figure 7A and B*). We found that HIGD1C interacts with CIV to alter the conformation of its enzymatic active center. In the absence of HIGD1A, co-overexpression of HIGD1C and COX4I2 increased oxygen sensitivity of CIV in HEK293T cells (*Figure 6E*), and overexpression of COX4I2 increased the stability of HIGD1C (*Figure 5—figure supplement 2B*). Since COX4I1, the ubiquitously expressed COX4 subunit, associates with HIGD1A in CIV assembly (*Timón-Gómez et al., 2020b*), the alternative COX4I2 subunit may assemble with HIGD1C. In opposition to its effect in *HIGD1A*-KO cells, COX4I2 overexpression in the presence of HIGD1A in WT cells decreases HIGD1C abundance, suggesting that HIGD1A and HIGD1C interact with CIV in the same domains (*Figure 5—figure supplement 1A*). These results indicate that coalitions of different CIV proteins may assemble under varying conditions and perform distinct physiological functions.

HIGD1C is evolutionarily closer to HIGD1A than to HIGD2A (*Timón-Gómez et al., 2020a*), which could explain why in normoxic conditions HIGD1C is able to substitute for HIGD1A function partially but not for HIGD2A (*Figure 5D and E*, *Figure 5—figure supplement 3E*). Unlike HIGD1A and HIGD2A, HIGD1C does not perform any apparent role in assembling the ETC complexes or supercomplexes (*Figure 5C*, *Figure 5—figure supplement 3D*). HIGD1C interacts with cytochrome *c* and CIV (*Figure 5A and B*) and, like HIGD1A, promotes CIV enzymatic activity in normoxia (*Figure 5D*). However, whereas HIGD1A is a positive regulator of CIV (*Hayashi et al., 2015*), our data indicate that HIGD1C serves as a negative modulator of CIV activity or is less efficient than HIGD1A in promoting CIV activity under limiting oxygen conditions (*Figure 6E*). Importantly, HIGD1C does not act in a CIV formed by standard subunits but in a CIV containing atypical tissue-specific isoforms known to be regulated by hypoxia, such as COX4I2, because increased sensitivity of the ETC to hypoxia is apparent only when both HIGD1C and COX4I2 are overexpressed (*Figure 6E and F*).

Our data allow us to conclude that the interaction of HIGD1C with hypoxic CIV containing atypical subunits results in an oxygen-sensing cytochrome *c* oxidase enzyme in CB glomus cells. However, while we demonstrated here that HIGD1C and COX4I2 are sufficient to confer oxygen sensitivity to CIV in HEK293T cells, additional components are likely to be required to fully reconstitute the oxygen sensitivity of CB glomus cells. Other proteins upregulated in glomus cells, such as the atypical CIV subunits NDUFA4L2 and COX8B and the glycolytic enzyme PCX (*Chang et al., 2015*; *Moreno-Domínguez et al., 2020*), are attractive candidates for further study to determine their potential contribution to CB oxygen sensing. The expression of three atypical CIV subunits in the CB correlates with the sufficiency of CIV alone to recapitulate the unusual oxygen dose response of the ETC in glomus cells (*Buckler and Turner, 2013*), suggesting that CIV is key for CB oxygen sensing. Future studies of oxygen consumption by mitochondria of glomus cells, when feasible, will further illuminate the roles of these proteins in CB oxygen sensing. While our study addresses CB oxygen sensing at the level of the oxygen sensor, how changes in ETC caused by HIGD1C and COX4I2 alter metabolic signaling by reactive oxygen species (ROS), lactate, and adenosine phosphates in hypoxia to regulate downstream G protein-coupled receptors and ion channels that stimulate neurotransmission remain to be elucidated (*Figure 7A*; *Chang, 2017*; *Evans, 2019*; *Holmes et al., 2018*; *Ortega-Sáenz and López-Barneo, 2020*).

CIV is the only ETC complex known to contain subunits that are tissue-specific and/or regulated by development, physiological changes (hypoxia and low glucose), and diseases (cancer, ischemia/reperfusion injury, and sepsis) (*Sinkler et al., 2017*; *Timón-Gómez et al., 2020a*). For example, COX4I2 is upregulated in hypoxia in general and promotes hypoxic pulmonary vasoconstriction in the lung (*Sinkler et al., 2017*; *Sommer et al., 2017*). *Higd1c* does not appear to be expressed in the lung at appreciable levels (*Figure 1—figure supplement 3C and D*), and the hypoxic response of the CB is faster than that of pulmonary arterial smooth muscle (seconds vs. minutes). In addition to the CB, *Higd1c* is expressed in kidney proximal tubules (*Suganthan et al., 2014*; *Figure 1—figure supplement 4G–J*). Compared to other nephron segments, the proximal tubules have the highest oxygen demand, exhibit greater ETC sensitivity to hypoxia, and are most susceptible to ischemia/reperfusion

injury (*Hall et al., 2009*). We speculate that HIGD1C modulates ETC activity and matches oxygen utilization to physiological function not only in the CB but in oxygen-sensitive cells in other organs. Due to imaging resolution limitations, our in situ hybridization results do not rule out the possibility that in the CB *Higd1c* is expressed in both glomus cells and sustentacular glial-like cells that ensheath them (*Figure 1D and E*), as these cell types are proposed to cooperate to promote sensory signaling (*Leonard et al., 2018*). Determining how HIGD1C and other atypical CIV proteins work together in the CB to mediate oxygen sensing will help us better understand how tissue- and condition-specific CIV subunits contribute to physiological function and disease and allow us to potentially target these proteins to treat diseases characterized by CB dysfunction.

# Materials and methods

## Mice
All animals were maintained in a barrier facility at 22–23°C with a 12 hr light/dark cycle and allowed ad libitum access to food and water. C57BL/6J (JAX) was used as the wild-type strain. Other mouse strains obtained from repositories were Th-Cre driver: B6.FVB(Cg)-*Tg(Th-cre)*$^{Fl172Gsat}$/Mmucd (MMRRC) (*Gong et al., 2007*) and ROSA-GCaMP3: B6;129S-*Gt(ROSA)26Sor*$^{tm38(CAG-GCaMP3)Hze}$/J (JAX) (*Zariwala et al., 2012*). Adult animals of both sexes from multiple litters were used in all experiments. *Higd1c* mutant strains were generated in this study by CRISPR/Cas9 gene editing. *Higd1c*$^{+/+}$ and *Higd1c*$^{-/-}$ animals were generated from crosses between *Higd1c*$^{+/-}$ parents. All experiments with animals were approved by the Institutional Animal Care and Use Committees at the University of California, San Francisco (AN183237-03), and the University of Calgary (AC16-0204).

## Human tissue
For human tissue, CB bifurcations were procured from research-consented, deidentified organ transplant donors through a collaboration with the UCSF VITAL Core (https://surgeryresearch.ucsf.edu/laboratories-research-centers/vital-core.aspx) and designated as non-human subjects research specimens by the UCSF IRB.

## Human cell lines and cell culture conditions
Human HEK293T embryonic kidney cells (CRL-3216, RRID:CVCL-0063) were obtained from ATCC. Cells were cultured in high-glucose Dulbecco's modified Eagle's medium (DMEM, Life Technologies) supplemented with 10% fetal bovine serum (FBS), 2 mM L-glutamine, 1 mM sodium pyruvate, and 50 mg/ml uridine at 37°C under 5% $CO_2$. Cell lines were routinely analyzed for mycoplasma contamination.

## Transgenic mice
*Higd1c* mutants were generated by injecting C57BL/6J embryos with in vitro transcribed sgRNA-1 and sgRNA-2 (10 ng/μl each) together with *Cas9* mRNA (50 ng/μl) and transferring injected embryos to pseudo-pregnant CD-1 females. Six founders were born and bred to C57BL/6J animals to isolate individual mutations transmitted through the germline, and sequences around sgRNA targets were PCR amplified and sequenced to identify mutations (*Figure 1—figure supplement 2A–C*). *Higd1c* mutant lines were maintained by breeding *Higd1c*$^{+/-}$ animals to each other.

## RNA purification and RT-qPCR
For mouse CB and kidney tissue, animals were anesthetized with isoflurane and decapitated, and tissues were dissected immediately. For all other tissues, animals were anesthetized and exsanguinated by perfusing PBS through the heart before decapitation and dissection. For human tissue, CB bifurcations were stored and transported in Belzer UW Cold Storage Solution (Bridge to Life) on ice. CBs were then dissected in UW Solution within 18 hr after harvest. After dissection, tissues were transferred to RNAprotect Tissue Reagent (QIAGEN) and stored at 4°C. For CB, kidney, adrenal gland, and all neuronal tissues, tissue pieces were disrupted and homogenized in a guanidine-isothiocyanate lysis buffer (Buffer RLT, QIAGEN) using a glass tissue grinder (Corning), followed by a 23-gauge needle and syringe, and purified by silica-membrane columns using the RNeasy Micro Kit (QIAGEN). For heart, liver, lung, and spleen, tissue pieces were ground using a glass tissue grinder in TRIzol (Invitrogen), and

RNA was purified by acid guanidinium thiocyanate-phenol-chloroform extraction followed by isopropanol precipitation. For cell culture, cells were pelleted and resuspended in Buffer RLT before RNA purification using columns. RNA quality was assessed by visualizing 28S and 18S rRNA by agarose gel electrophoresis, and RNA concentration was measured with a Nanodrop ND-1000 Spectrophotometer (Thermo). RNA was stored at –80°C.

Two-step RT-qPCR was performed. First, purified total RNA was synthesized into cDNA:RNA hybrids with Maxima H Minus Reverse Transcriptase (Thermo) and primed using equal amounts of oligo(dT)15 primers (Promega) and random hexamers (Thermo). RNasin Plus RNase Inhibitor (Promega) was also added to the mixture. Next, qPCR was performed using PowerUp SYBR Green Master Mix (Applied Biosystems), at 10 µl reaction volume, following the manufacturer's instructions. Three technical replicates were performed for each reaction and plated in TempPlate 384-well PCR plates (USA Scientific). Sample plates were run using a QuantStudio 5 Real-Time PCR System (Applied Biosystems) using a 40-cycle amplification protocol.

QuantStudio software was used to calculate threshold cycle (Ct) values. Undetermined Ct values were set to Ct = 40. Ct values were averaged for all technical replicates for each biological sample and normalized to either *Actb* or to *GAPDH*, using the ΔCt method.

## BaseScope in situ hybridization

Animals were anesthetized with isoflurane, decapitated, and dissected. Tissue was fixed in RNase-free 4% PFA/PBS overnight at 4°C and equilibrated serially in 10% sucrose/PBS for >1 hr, 20% sucrose/PBS for >2 hr, and 30% sucrose/PBS overnight, all at 4°C. Tissue was then embedded in O.C.T. (TissueTek) and stored at –80°C. The tissue was sectioned at 10 µm using a Leica CM3050S cryostat and stored at –80°C.

Following the BaseScope protocol for fixed frozen sections, slides were baked for 50 min at 60°C and post-fixed with 10% neutral-buffered saline for 15 min at 60°C. This was followed by target retrieval for 5 min at 100°C and protease III treatment for 30 min at 40°C. Using the BaseScope Duplex Detection Reagent kit (Advanced Cell Diagnostics, 323810), subsequent steps of hybridization and detection followed the vendor's protocol. Probes are listed in *Table 2*. The probe set for *Higd1c* was custom-designed to target only the first two exons of *Higd1c*. For detection of *Th* mRNA, amplification steps 7 and 8 were reduced from 30 min and 15 min, respectively, to 15 min and 7.5 min for some samples. Images were collected on a Nikon Ti widefield inverted microscope using a DS-Ri2 color camera. Two sets of experiments were performed on tissues from C57BL/6J (2), *Higd1c*^{+/+} (3), and *Higd1c*^{-/-} (2) animals (*Figure 1D and E*, *Figure 1—figure supplement 4A–E,G–J*).

## Immunostaining

Cultured cells on coverslips were fixed with 1% or 4% PFA/PBS for 10 min at 22°C and used immediately or stored in PBS at 4°C. Tissue was fixed in 4% PFA/PBS for 10 min at 22°C and equilibrated in 30% sucrose overnight at 4°C. Tissue was embedded in O.C.T. (TissueTek) and stored at –80°C. Sections were cut at 10 µm using a Leica CM3050S cryostat and stored at –80°C. Fixed cells or tissue sections were incubated with primary antibodies overnight at 4°C. Primary antibodies were mouse anti-DDK/FLAG, mouse anti-HSP60, rabbit anti-TH, and rat anti-CD31, all used at 1:500. For kidney sections, fluorescein-labeled *Lotus tetragonolobus* lectin (LTL) was added during the primary antibody treatment. Incubation with secondary antibodies (1:250) conjugated to either Alexa Fluor 488, Alexa Fluor 555 (Life Technologies), or Cy3 (Jackson ImmunoResearch) was 45 min at 22°C. Samples were then incubated with DAPI (1 ng/ml, Life Technologies) for 5 min at 22°C and mounted in Mowiol 4-88 (Polysciences) with DABCO (25 mg/ml, Sigma-Aldrich). Samples were imaged using a Leica SPE confocal microscope for cell culture and a Zeiss Axio Observer D1 widefield inverted microscope for tissue sections. Quantification of TH-positive cells was performed on 1/3 of the total CB using 1 of 3 sets of adjacent sections (*Figure 1G–I*).

## Whole-body plethysmography

Adult mice were removed from the housing room and placed in the procedure room for a minimum of 1 hr before starting the experiment to acclimate. Ventilation of unanesthetized, awake mice was measured using a commercial system for whole-body plethysmography (Scireq). Chamber pressure was detected by a pressure transducer and temperature and humidity by a sensor. These signals were

**Table 2.** Reagents and resources.

| Reagent or resource | Source | Identifier |
|---|---|---|
| **Mouse strains** | | |
| C57BL/6J (wild-type) | JAX | 000664 |
| B6.FVB(Cg)-*Tg(Th-cre)*$^{Fl172Gsat}$/Mmucd | MMRRC | 031029-UCD |
| B6;129S-*Gt(ROSA)26Sor*$^{tm38(CAG-GCaMP3)Hze}$/J | JAX | 014538 |
| B6.*Higd1c 3-1* | This paper | |
| B6.*Higd1c 1-1* | This paper | |
| B6.*Higd1c 5-3* | This paper | |
| | | |
| **sgRNA primers (target sequence in bold)** | | |
| sgRNA-1 (forward): TAATACGACTCACTATA**GGGAGTCTCTCGATTTCCGG**GTTTTAGAGCTAGAA | This paper | |
| sgRNA-2 (forward): TAATACGACTCACTATAGG**CTGATTTAAGGAGTGAGTGC**GTTTTAGAGCTAGAA | This paper | |
| sgRNA (reverse, common): AAAAAAAGCACCGACTCGGTGCCACTTTTTCAAGTTGATAACGGACTAGCCTTATTTTAACTTGCTATTTCTAGCTCTAAAAC | This paper | |
| | | |
| **Genotyping primers** | | |
| *Higd1c*-P8: GTCAGGTGGCCCCTGATGAAA | This paper | |
| *Higd1c*-P9: GTGCACGAGCAGACTGGTTCT | This paper | |
| *Higd1c*-P11: GGATATCACAGCCACAGAGGACG | This paper | |
| | | |
| **Mouse qPCR primers** | | |
| *Actb*-F: AGCCATGTACGTAGCCATCC | This paper | |
| *Actb*-R: GCCATCTCTTGCTCGAAGTC | This paper | |
| *Higd1c* (5 exon)-F: CACGTACAAGGGCTGCATGG | This paper | |
| *Higd1c* (5 exon)-R: ACCTAGAGTCACGGCTCCC | This paper | |
| *Higd1c* (4 exon)-F: CCAGCACGTACAAGAGAGAAA | This paper | |
| *Higd1c* (4 exon)-R: ACGTGGATGAGATGAAGGGAC | This paper | |
| | | |
| **Rat qPCR primers** | | |
| *GADPH*-F: CAAGTTCAACGGCACAGTCAAG | *Kim et al., 2011* | |
| *GADPH*-R: ACATACTCAGCACCAGCATCAC | *Kim et al., 2011* | |
| *Higd1c*-F1: CCTGTGCTGATCAAAGAGCA | This paper | |
| *Higd1c*-R1: CTGACCACTCATCTGAAGAC | This paper | |
| *Higd1c*-R2: CTGCTGACCACTCATCTGAA | This paper | |
| *Kcnk3*-F: GCAGAAGCCGCAGGAGTTC | *Kim et al., 2011* | |
| *Kcnk3*-R: GCCCGCACAGTTGGAGATTTAG | *Kim et al., 2011* | |
| *Kcnk9*-F: CGGTGCCTTCCTCAATCTTGTG | *Kim et al., 2011* | |
| *Kcnk9*-R: TGGTGCCTCTTGCGACTCTG | *Kim et al., 2011* | |
| *Th*-F: TCGGAAGCTGATTGCAGAGA | *Feng et al., 2020* | |

*Table 2 continued on next page*

*Table 2 continued*

| Reagent or resource | Source | Identifier |
|---|---|---|
| *Th*-R: TTCCGCTGTGTATTCCACATG | *Feng et al., 2020* | |
| *Olr59*-F: TCATTCACGCTCTCTCAGCA | *von der Weid et al., 2015* | |
| *Olr59*-R: CCATGCCGATTTGGACTGTT | *von der Weid et al., 2015* | |
| **Human qPCR primers** | | |
| *GADPH*-F: ACCACAGTCCATGCCATCAC | *Maßberg et al., 2016* | |
| *GADPH*-R: TCCCACCACCCTGTTGCTGTA | *Maßberg et al., 2016* | |
| *HIGD1A*-F1: CAACAGACACAGGTGTTTCC | This paper | |
| *HIGD1A*-R1: CAATTGCTGCAAAACCCGCT | This paper | |
| *HIGD2A*-F: GCCCCACTGTTTACAGGAAT | This paper | |
| *HIGD2A*-R: GCGCATCATGAGCTGAGAG | This paper | |
| *HIGD1C*-F: GAAGGCCAATTATCCCGACT | This paper | |
| *HIGD1C*-R: GCTTGTAAAGACCACAGGAC | This paper | |
| *COX4I1*-F: CAAGCGAGCAATTTCCACCT | This paper | |
| *COX4I1*-R: CCTTCTCCTTCAATGCCTTC | This paper | |
| *COX4I2*-F: GAGGGATGCACAGCTCAGAA | This paper | |
| *COX4I2*-R: CTTCTCCTTCTCCTTCAGGG | This paper | |
| *NDUFA4L2*-F: GATGATCGGCTTAATCTGCC | This paper | |
| *NDUFA4L2*-R: GTATTGGTCATTGGGGCTCA | This paper | |
| *TH*-F: GCTGGACAAGTGTCATCACCTG | OriGene | HP234519 |
| *TH*-R: CCTGTACTGGAAGGCGATCTCA | OriGene | HP234519 |
| *OR51E2*-F2: TCATCCCATTGTGCGTGTTG | This paper | |
| *OR51E2*-R2: CACCCGTGTTCTGATCTGTTTG | This paper | |
| **BaseScope in situ hybridization probes** | | |
| BA-Mm-*Higd1c*-2zz-st | ACD | 862241 |
| BA-Mm-*Th*-3EJ-C2 | ACD | 854771-C2 |
| *dapB*-1ZZ-C1/*dapB*-1ZZ-C2 | ACD | 700141 |
| Mm-*Ppib*-1ZZ | ACD | 701081 |
| Mm-*Polr2a*-1ZZ-C2 | ACD | 701101-C2 |
| **Plasmids** | | |
| *HIGD1C*-Myc-FLAG in pCMV6-Entry (human) | OriGene | RC225015 |
| *Higd1c*-Myc-FLAG in pCMV6-Entry (mouse) | OriGene | MR220387 |
| *COX4I1*-Myc-FLAG in pCMV6-Entry | OriGene | RC209374 |
| *COX4I2*-Myc-FLAG in pCMV6-Entry | OriGene | RC209204 |
| pCMV6-A-Entry-Hygro | OriGene | PS100024 |
| pCMV6-A-Entry-BSD | OriGene | PS100022 |

*Table 2 continued on next page*

*Table 2 continued*

| Reagent or resource | Source | Identifier |
|---|---|---|
| *HIGD1A*-Myc-FLAG in pCMV6-A-Entry-Hygro | *Timón-Gómez et al., 2020b* | |
| *HIGD2A*-Myc-FLAG in pCMV6-A-Entry-Hygro | *Timón-Gómez et al., 2020b* | |
| *HIGD1C*-Myc-FLAG in pCMV6-A-Entry-Hygro (human) | This paper | |
| *Higd1c*-Myc-FLAG in pCMV6-A-Entry-Hygro (mouse) | This paper | |
| *COX4I1*-Myc-FLAG in pCMV6-A-Entry-BSD | This paper | |
| *COX4I2*-Myc-FLAG in pCMV6-A-Entry-BSD | This paper | |

| Primary antibodies/stains | | |
|---|---|---|
| Mouse anti-DDK/FLAG | OriGene | TA50011 |
| Mouse anti-HSP60 | ECM Biosciences | HM-4381 |
| Rabbit anti-TH | Abcam | ab112 |
| Rat anti-CD31 | BD Pharmingen | 553370 |
| *Lotus tetragonolobus* lectin-Fluorescein | Vector Labs | FL-1321 |
| Mouse anti-ATP5A | Abcam | ab14748 |
| Mouse anti-β-ACTIN | Abcam | ab8227 |
| Mouse anti-CORE2 | Abcam | ab8227 |
| Mouse anti-COX1 | Abcam | ab14705 |
| Mouse anti-COX4I1 | Abcam | ab14744 |
| Rabbit anti-COX4I2 | Abnova | H00084701-M01 |
| Mouse anti-COX5B | Santa Cruz | sc-374417 |
| Mouse anti-β-TUBULIN | Sigma-Aldrich | C4585 |
| Rabbit anti-UQCRB | Abcam | ab122837 |
| Mouse anti-NDUFA9 | Abcam | ab14713 |
| Rabbit anti-NDUFB11 | Abcam | ab183716 |
| Rabbit anti-SDHA | Abcam | ab14715 |
| Mouse anti-cytochrome *c* | Santa Cruz | sc-13156 |

integrated using IOX2 software (Scireq) to calculate the instantaneous flow rate. Baseline breathing was established during a period of at least 30 min in control gas. The baseline was followed by two hypoxic periods and one hypercapnic period, each lasting 5 min, interspersed with recovery periods of 10 min in control gas (*Figure 2—figure supplement 1A–C*). Gas mixtures for control, hypoxia, and hypercapnia were 21% $O_2$/79% $N_2$, 10% $O_2$/90% $N_2$, and 5% $CO_2$/21% $O_2$/79% $N_2$, respectively (Airgas). The flow rate was held constant at 1.5 l/min by a flowmeter.

Breathing traces were collected, and ventilatory parameters were calculated by IOX2 software (Scireq). Breath inclusion criteria were set in the software to the following: (1) inspiratory time (0.07–1 s), (2) expiratory time (0.1–1 s), (3) tidal volume (0.05–0.8 ml), and (4) respiratory rate (10–320 breaths/ml). Data for all accepted breaths were exported and processed using a custom R script to calculate the average respiratory rate, tidal volume, and minute ventilation for each period. Trials were rejected if many accepted breaths occurred during periods of animal sniffing, grooming, or movement; we used a respiratory rate of 215 breaths/ml as a cutoff for inclusion of trials. A trial was rejected if any of the normoxic periods had an average respiratory rate that exceeded 215 breath/min (comparable to the mean of wild-type in hypoxia) in order to assess calm breathing and exclude artifacts from sniffing, grooming, and movement that correlated with high-frequency events above the ventilation in hypoxia. If a trial was rejected, the animal was retested on subsequent days, for up to four trials,

until stable ventilation was reached in control normoxic periods. Data collection and analysis were automated using above inclusion/exclusion criteria. The percentage of experiments rejected between *Higd1c*$^{+/+}$ and *Higd1c*$^{-/-}$ animals by allele were not statistically significant by the *Z*-test of proportions (p=0.7039, 0.5353, and 0.7114 for *1-1*, *3-1*, and *5-3* alleles, respectively). Because the effect size could not be estimated and variance for *Higd1c* animals was unknown, no sample size determination was performed, but sample sizes were comparable to other published studies (*Del Rio et al., 2013*; *Moreno-Domínguez et al., 2020*; *Soliz et al., 2005*).

We did not normalize tidal volume or minute ventilation to body weight because for the animals used in our study body weight did not correlate well with respiratory rate, tidal volume, or minute ventilation in wild-type or mutant animals in normoxia or hypoxia. Body weights of *Higd1c*$^{+/+}$ and *Higd1c*$^{-/-}$ animals were not significantly different by two-way ANOVA with Sidak correction (p=0.9905, 0.9255, and 0.9562 for *1-1*, *3-1*, and *5-3* alleles, respectively). Nevertheless, we verified that all differences that were statistically significant without normalizing by body weight were also significant if we normalized to body weight (p<0.05). In addition to comparing mean values of ventilatory parameters, we showed the % change in the ventilatory parameters for each animal before and after hypoxia ('Hypoxic response'), using each animal as its own control.

## Carotid sinus nerve recordings

Animals were heavily anesthetized with isoflurane and then decapitated (lower cervical level). The carotid bifurcation, including the CB, carotid sinus nerve (CSN), and superior cervical ganglion, was quickly isolated en bloc for in vitro perfusion as described previously (*Roy et al., 2012*). The carotid bifurcation was then transferred to a dissection dish containing physiological saline (115 mM NaCl, 4 mM KCl, 24 mM NaHCO$_3$, 2 mM CaCl$_2$, 1.25 mM NaH$_2$PO$_4$, 1 mM MgSO$_4$, 10 mM glucose, 12 mM sucrose) bubbling 95% O$_2$/5% CO$_2$. After 15–20 min, the isolated tissue was transferred to a recording chamber (AR; custom-made) with a built-in waterfed heating circuit, and the common carotid artery was immediately cannulated for luminal perfusion with physiological saline equilibrated with 100 mmHg PO$_2$ and 35 mmHg PCO$_2$ (balance N$_2$). After gross dissection, connective tissue was removed, and the CSN was carefully desheathed. The carotid sinus region was bisected. The occipital and internal and external carotid arteries were ligated, and small incisions were made on the internal and external carotid arteries to allow perfusate to exit. A peristaltic pump was used to set the perfusion rate at 7 ml/min, which was sufficient to maintain a constant pressure of 90–100 mmHg at the tip of the cannula. The perfusate was equilibrated with computer-controlled gas mixtures using CO$_2$ and O$_2$ gas analyzers (CA-2A and PA1B, Sable Systems); a gas mixture of 100 mmHg PO$_2$ and 35 mmHg PCO$_2$ (balance N$_2$) was used to start the experiments (yielding pH ~ 7.4). Before reaching the cannula, the perfusate was passed through a bubble trap and heat exchanger. The temperature of the perfusate was measured continuously as it departed the preparation and maintained at 37 ± 0.5°C. The effluent from the chamber was recirculated.

Chemosensory discharge was recorded extracellularly from the whole desheathed CSN, which was placed on a platinum electrode and lifted into a thin film of paraffin oil. A reference electrode was placed close to the bifurcation. CSN activity was monitored using a differential AC amplifier (Model 1700, A-M Systems) and a secondary amplifier (Model 440, Brownlee Precision). The neural activity was amplified, filtered (0.3–1 kHz), displayed on an oscilloscope, rectified, integrated (200 ms time constant), and stored on a computer using an analog-to-digital board (Digidata 1322A, Axon Instruments) and data acquisition software (Axoscope 9.0). Recording was only attempted in nerves that survived cleaning and desheathing. The presence of action potentials under baseline conditions was used as the only test of preparation viability; data was obtained from all preparations deemed viable according to this criterion.

The following protocol was used for all experiments: (1) the CB was perfused for 5 min with normoxia (100 mmHg PO$_2$/35 mmHg PCO$_2$) to determine baseline CSN activity; (2) neural responses were obtained by challenging the CB for 4 min with mild, moderate, and severe hypoxia (80, 60, and 40 mmHg PO$_2$, respectively) interspersed with normoxia; and (3) a hypercapnic (60 mmHg PCO$_2$) challenge was given for 4 min (*Figure 3A and B*).

Data were analyzed offline using custom software (*Wilson, 2022*). CSN activity was divided into 60 s time bins, and the activity in each bin was rectified and summed (expressed as integrated neural discharge). The neural responses for different conditions in the protocol were normalized to the

baseline (normoxic) condition. Data acquisition and CSN activity analysis were performed blinded to genotype.

## Calcium imaging

*Th-Cre^{Tg/+}*; *ROSA-GCaMP3^{Tg/Tg}*; *Higd1c^{+/+}* and *Th-Cre^{Tg/+}*; *ROSA-GCaMP3^{Tg/Tg}*; *Higd1c^{-/-}* animals expressing GCaMP3 in glomus cells were generated, and CB was imaged as previously described (*Chang et al., 2015*). Animals were anesthetized with isoflurane and decapitated. Carotid bifurcations were dissected and cleaned in PBS to keep only the CB attached to the bifurcation. The preparation was then incubated in a physiological buffer (115 mM NaCl, 5 mM KCl, 24 mM $NaHCO_3$, 2 mM $CaCl_2$, 1 mM $MgCl_2$, 11 mM glucose) at 26°C in a tissue culture incubator with 5% $CO_2$ before transfer to the recording chamber for imaging.

At baseline, the CB was superfused by gravity at 5 ml/min with physiological buffer bubbling 95% $O_2$/5% $CO_2$ in the reservoir to maintain $PO_2$ ~ 700 mmHg and pH 7.4 in the imaging chamber at 22°C. Buffer pH was lowered to 6.8 by reducing $NaHCO_3$ to 7 mM with equimolar substitution of NaCl while bubbling 95% $O_2$/5% $CO_2$. Two levels of hypoxia at $PO_2$ ~ 25 mmHg and 50 mmHg were generated by bubbling physiological buffer in the reservoir with 90% $N_2$/5% $O_2$/5% $CO_2$ and 95% $N_2$/5% $CO_2$, respectively. The preparation was sequentially stimulated with low pH and hypoxia for periods of 4.5 min each, with 3 min of recovery between stimuli. These were followed by KCl (40 mM) and CN (1 mM) for periods of 2.25 min each, with 4.5 min of recovery between stimuli (*Figure 3E and F*).

Imaging was performed on a Zeiss LSM 7 MP two-photon microscope with a Coherent Ultra II Chameleon laser and a sensitive gallium arsenide phosphide (GaAsP) detector. Preparations were excited at 960 nm, and emission was collected at 500–550 nm. Using a ×20 water immersion objective, we acquired Z-stacks at 2 μm intervals at a resolution of 1024 × 1024 pixels and up to 60–85 μm of tissue depth.

Regions of interest (ROIs) corresponding to individual glomus cells were identified and analyzed in ImageJ. All ROIs were included in the data. Fpre fluorescence was defined as the average fluorescence over the four frames immediately prior to the onset of the stimulus in the chamber. Mean and peak fluorescence were calculated over the duration when the stimulus was present in the imaging chamber. The ratio of Fstim/Fpre was calculated by dividing the mean and peak by Fpre just preceding the stimulus. Data acquisition and ROI analysis were carried out blinded to genotype.

## Metabolic imaging

*Higd1c^{+/+}* and *Higd1c^{-/-}* animals were anesthetized with isoflurane and decapitated. Carotid bifurcations were dissected and cleaned in PBS to keep only the CB attached to the bifurcation. The preparation was then incubated in 50 μg/ml rhodamine 123 (Thermo Fisher) in a physiological buffer (115 mM NaCl, 5 mM KCl, 24 mM $NaHCO_3$, 2 mM $CaCl_2$, 1 mM $MgCl_2$, 11 mM glucose) at 26°C in a tissue culture incubator with 5% $CO_2$ for 30 min before transfer to the recording chamber for imaging.

At baseline, the CB was superfused by gravity at 5 ml/min with physiological buffer bubbling 95% $O_2$/5% $CO_2$ in the reservoir to maintain $PO_2$ ~ 700 mmHg and pH 7.4 in the imaging chamber at 22°C. Hypoxia down to $PO_2$ ~ 10 mmHg was generated by bubbling physiological buffer in the reservoir with 95% $N_2$/5% $CO_2$ for 7.5 min. $PO_2$ was measured using a Clark style oxygen sensor (OX-50, Unisense). After 6 min of baseline recording, the preparation was stimulated with hypoxia for a period of 7.5 min followed by 7.5 min of recovery between stimuli (*Figure 4—figure supplement 1*). This was followed by CN (1 mM) and FCCP (2 μM) for periods of 2.25 min each, with 7.5 min of recovery between stimuli. For control experiments with a single FCCP stimulus, the protocol was as follows: 6 min physiological buffer, 2.25 min FCCP (2 μM), and 6 min physiological buffer, all in buffer bubbled with 95% $N_2$/5% $CO_2$. Imaging and analysis methods were the same for both experimental protocols.

Imaging was performed on a Zeiss LSM 7 MP two-photon microscope with a Coherent Ultra II Chameleon laser and a sensitive gallium arsenide phosphide (GaAsP) detector. Preparations were excited at 960 nm, and emission was collected at 500–550 nm. Using a ×20 water immersion objective, we acquired Z-stacks at 2 μm intervals at a resolution of 1024 × 1024 pixels and up to 60–85 μm of tissue depth.

ROIs corresponding to individual glomus cells were identified and analyzed in ImageJ. All ROIs were included in the data except as indicated in specific analyses. We performed baseline subtraction after linear interpolation to account for a linear decrease in baseline fluorescence occurring over the

time course of the experiment. First, the fluorescence trace of each ROI was smoothed using a three-point centered rolling average, and the baseline was calculated using linear interpolation between the inter-stimulus intervals. This baseline was then subtracted from the original traces. Fpre fluorescence was defined as the fluorescence immediately prior to the onset of the stimulus in the chamber. Mean and peak fluorescence were calculated over the duration when the stimulus was present in the imaging chamber. The ratio of Fstim/Fpre was calculated by dividing the mean and peak by Fpre just preceding the stimulus. Data acquisition and ROI analysis were carried out blinded to genotype.

### Stable cell line construction

*HIGD1C*-Myc-DDK/FLAG constructs in pCMV6-Entry were cloned under the control of a CMV promoter in the pCMV6-A-Entry-Hygro plasmid, and COX4I1/COX4I2-Myc-DDK constructs in pCMV6-Entry were cloned in the pCMV6-A-Entry-BSD, using Sfg1 and Pme1 sites. 1–2 μg of vector DNA was mixed with 5 μl of Lipofectamine (Thermo Fisher) in OPTIMEM-I media (GIBCO) to transfect $1.5 \times 10^6$ cells according to the manufacturer's instructions. After 48 hr, media was supplemented with 200 μg/ml of hygromycin or 10 μg/ml of blasticidin and maintained for at least 21 days.

### Cell culture experimental conditions

*HIGD1A*-KO and *HIGD2A*-KO cells were constructed in HEK293T using the TALEN technology as described in *Timón-Gómez et al., 2020b*. HEK293T cells were grown in 25 mM glucose DMEM (Life Technologies) supplemented with 10% FBS, 2 mM L-glutamine, 1 mM sodium pyruvate, and 50 μg/ml uridine without antibiotics, at 37°C under 5% $CO_2$. For metabolic imaging involving HEK293T cells, 10 mm glass coverslips were placed into 1.96 $cm^2$ wells and coated with 0.2 mg/ml poly-D-lysine for at least 2 hr at room temperature (RT). Two days before the experiment, $7.5 \times 10^4$ cells in 500 μl of cell media were seeded into each well. For hypoxia experiments, cell cultures were exposed to 1% $O_2$ for up to 24 hr, or as controls, to standard cell culture oxygen tension (18.6% $O_2$). Experiments under controlled oxygen tensions were performed in a HypOxystation H35 (HypOxygen) to minimize undesired oxygen reperfusion. Routinely, cells were analyzed for mycoplasma contamination.

### Mitochondrial biochemistry

Mitochondrial fractions were obtained as previously described in *Bourens et al., 2014*; *Fernández-Vizarra et al., 2010*; *Timón-Gómez et al., 2020b* from ten 80% confluent 15 cm plates or from 1 l of liquid culture. Whole-cell extracts were obtained from pelleted cells solubilized in RIPA buffer (25 mM Tris–HCl [pH 7.6], 150 mM NaCl, 1% NP-40, 1% sodium deoxycholate, and 0.1% SDS) with 1 mM PMSF for 20 min. Extracts were cleared after 5 min centrifugation at 15,000 rpm at 4°C.

Proteins were extracted from purified mitochondria in native conditions with either digitonin at a proportion of 1:2 of protein or with n-dodecyl-β-D-maltoside (DDM) at a concentration of 0.4%. Samples were incubated on ice for 10 min and pelleted at $10,000 \times g$ for 30 min at 4°C. Samples were prepared for Blue Native Electrophoresis and/or Complex I and Complex IV *in-gel* activity (IGA) assays as described (*Timón-Gómez et al., 2020c*). Immunoprecipitation of HIGD1C-Myc-DDK-tagged proteins was performed using 1 mg of mitochondria, extracted in 1.5 M aminocaproic acid, 50 mM Bis-Tris pH 7, 1% digitonin, 1 mM PMSF, and 8 μl of protease inhibitor cocktail (Sigma, P8340) for 10 min on ice. Samples were pelleted at $10,000 \times g$ for 30 min at 4°C, and the extract (Ex) was incubated for 2–3 hr at 4°C with 30 μl of FLAG-conjugated beads (anti-DYDDDDK beads, Sigma) or empty beads (Thermo Scientific), previously washed in PBS. Beads were washed five times in 1 ml of 1.5 M aminocaproic acid, 50 mM Bis-Tris (pH 7), 0.1% digitonin, and boiled for 5 min with 50 μl of Laemmli buffer two times to release bound material. Representative amounts of all fractions were loaded on 14% SDS-PAGE gels.

### Complex specific assay and oxygen consumption rate

Mitochondrial respiratory chain CIV activity was performed according to established methods (*Barrientos et al., 2009*). Citrate synthase activity was used as a control. Enzymatic activities were expressed relative to the total amount of extracted protein.

OCR in normoxia was measured polarographically using a Clark-type electrode from Hansatech Instruments (Norfolk, UK) at 37°C. Approximately $2 \times 10^6$ cells were trypsinized and washed with PBS, and then resuspended in 0.5 ml of permeabilized-cell respiration buffer (PRB) containing 0.3 M

mannitol, 10 mM KCl, 5 mM MgCl$_2$, 0.5 mM EDTA, 0.5 mM EGTA, 1 mg/ml BSA, and 10 mM KH$_2$PO$_4$ (pH 7.4) at 37°C, supplemented with 10 units of hexokinase. The cell suspension was immediately placed into the polarographic chamber to measure endogenous respiration. Digitonin permeabilization (0.02 mg/ml) was performed to assay substrate-driven respiration, using FADH-linked substrates (10 mM succinate plus 5 mM glycerol-3-phosphate) in the presence of 2.5 mM ADP (phosphorylation state). Oligomycin-driven ATP synthesis inhibition (0.75 µg/ml) was assayed to obtain the non-phosphorylating state. Maximal oxygen consumption was reached by successive addition of the uncoupler CCCP (up to 0.4 µM). 0.8 µM KCN was used to assess the mitochondrial specificity of the oxygen consumption measured, and values were normalized by total cell number.

High-resolution respirometry was used to determine mitochondrial oxygen consumption and ascorbate/TMPD-dependent respiration in normoxia and hypoxia. Measurements were performed in intact or digitonin-permeabilized cells, respectively, in an Oxygraph-2k (Oroboros Instruments, Austria). Assays were performed according to manufacturer's SUIT protocols, using 2–4 × 10$^5$ cells washed with PBS and resuspended in Mir05 medium, and results were normalized by cell number. To analyze oxygen kinetics and determine the apparent Km (p50$_{mito}$) for oxygen, we used the software Datlab 2 (Oroboros Instruments) by integrating a hyperbolic function of mitochondrial oxygen consumption and oxygen pressure during and transition from aerobic respiration to anoxia (*Gnaiger, 2001*; *Gnaiger et al., 1995*). For this purpose, intact cell respiration experiments were performed using 2 × 10$^6$ cells in a respiration medium containing 0.5 mM EGTA, 3 mM MgCl$_2$, 60 mM K-lactobionate, 20 mM taurine, 10 mM KH$_2$PO$_4$, 20 mM HEPES, 110 mM sucrose, and 1 mg/ml BSA.

## Extraction of total mitochondrial cytochromes

8 mg of mitochondria were extracted with 330 mM KCl, 50 mM Tris–HCl (pH 7.5), and 10% potassium deoxycholate. Samples were mixed by inversion three times and pelleted for 15 min at 40,000 × *g* at 4°C. The clear supernatant was transferred to a new tube, and a final concentration of 2% potassium cholate was added. The extract was divided into two equal aliquots in 1 ml quartz cuvettes, and the baseline was established. Then, the reference aliquot was oxidized with potassium ferricyanide, and the other was reduced with a few grains of sodium dithionite. Differential reduced vs. oxidized spectrum was recorded from 450 to 650 nm.

## Statistical analysis

Data analysis and statistical tests were performed using Microsoft Excel, custom-written scripts in R, and GraphPad Prism software. All data are biological replicates. Group data were analyzed by the Shapiro–Wilk test to determine whether the data was normally distributed with the critical *W* value set at a 5% significance level. Normally distributed data are presented as mean ± standard error of the mean (SEM) and compared by one-way analysis of variance (ANOVA) followed by Tukey's test or Holm–Sidak correction, two-way ANOVA followed by Tukey's test for all pairwise comparisons or Sidak correction for multiple pairwise comparisons, or Dunnett's test for multiple comparisons to a single group. For comparisons that included groups that did not fit the assumption of normal distribution, data are presented as median and interquartile interval and compared by Mann–Whitney *U*-tests followed by Holm–Sidak correction for multiple comparisons or Kruskal–Wallis test with Dunn's test for multiple comparisons to a single group. For whole-body plethysmography experiments, 2/54 (hypoxia) and 7/54 (hypercapnia) data groups were not normally distributed ($p<0.05$ by Shapiro–Wilk test). All significant differences found by parametric statistical tests were significant using nonparametric tests. The *Z*-test of proportions was used to compare the proportion of glomus cells with responses at different thresholds. Chi-square test was performed for analysis of Mendelian inheritance of *Higd1c* alleles and to determine whether the distributions of glomus cells responsive to different hypoxic stimuli were drawn from the same population. All tests were two-tailed. No statistical method was used to predetermine sample size.

## Materials availability

All unique/stable reagents generated in this article (plasmids, cell line, and mice) are available upon request from Andy Chang (Andy.Chang@ucsf.edu) and Antoni Barrientos (abarrientos@med.miami.edu) with a completed Materials Transfer agreement. Custom scripts used for analysis are available as source code or in a public repository (*Wilson, 2022*).

## Acknowledgements

We thank Peter Lee for technical assistance, Alex Diaz de Arce for assistance with RNAseq analysis, Kailyss Freeman and the UCSF VITAL Core for coordinating donor tissue, ACD Bio for BaseScope in situ hybridization, Blair Gainous, Pauline Colombier, Brian Black, and the CVRI Transgenic Mouse Core for guidance and technical support in generating *Higd1c* mutant mice, Chris Allen for use of his two-photon microscope and cryostat, Jeremy Reiter for use of his widefield microscope, and the UCSF Nikon Imaging Center for use of their confocal and widefield microscopes. We sincerely thank Donor Network West, and most importantly the organ donors and their families, who give this precious gift to further scientific research.

## Additional information

### Funding

| Funder | Grant reference number | Author |
| --- | --- | --- |
| Muscular Dystrophy Association | Career Development Award 862896 | Alba Timón-Gómez |
| National Institutes of Health | UCSF Transplant T32 FAVOR Grant P0548805 | Alexandra L Scharr |
| University of California, San Francisco | Physician-Scientist Scholars Program | James M Gardner |
| Canadian Institutes of Health Research | Research Grant 201603PJT/366421 | Richard JA Wilson |
| Alberta Innovates - Health Solutions | Senior Scholar | Richard JA Wilson |
| National Institute of General Medical Sciences | R35 Grant GM118141 | Antoni Barrientos |
| University of California, San Francisco | Sandler Program for Breakthrough Biomedical Research New Frontier Award | Andy J Chang |
| University of California, San Francisco | Cardiovascular Research Institute | Andy J Chang |

The funders had no role in study design, data collection and interpretation, or the decision to submit the work for publication.

### Author contributions

Alba Timón-Gómez, Data curation, Formal analysis, Validation, Investigation, Visualization, Methodology, Writing - original draft, Writing - review and editing; Alexandra L Scharr, Nicholas Y Wong, Data curation, Software, Formal analysis, Validation, Investigation, Visualization, Methodology, Writing - original draft, Writing - review and editing; Erwin Ni, Formal analysis, Validation, Investigation, Visualization, Methodology; Arijit Roy, Data curation, Formal analysis, Validation, Investigation, Visualization, Methodology, Writing - original draft; Min Liu, Validation, Investigation, Methodology, Writing - original draft; Julisia Chau, Validation, Investigation, Methodology; Jack L Lampert, Homza Hireed, Formal analysis, Validation, Investigation, Methodology; Noah S Kim, Investigation, Visualization, Methodology; Masood Jan, Investigation, Visualization; Alexander R Gupta, Ryan W Day, Resources; James M Gardner, Resources, Supervision, Methodology; Richard JA Wilson, Software, Supervision, Funding acquisition, Validation, Investigation, Methodology, Writing - review and editing; Antoni Barrientos, Resources, Supervision, Funding acquisition, Validation, Investigation, Visualization, Methodology, Writing - original draft, Project administration, Writing - review and editing; Andy J Chang, Conceptualization, Resources, Data curation, Formal analysis, Supervision, Funding acquisition, Validation, Investigation, Visualization, Methodology, Writing - original draft, Project administration, Writing - review and editing

## Author ORCIDs
Jack L Lampert ![ORCID] http://orcid.org/0000-0002-5367-7707
Richard JA Wilson ![ORCID] http://orcid.org/0000-0001-9942-4775
Antoni Barrientos ![ORCID] http://orcid.org/0000-0001-9018-3231
Andy J Chang ![ORCID] http://orcid.org/0000-0002-1247-4794

## Ethics

Human subjects: For human tissue, CB bifurcations were procured from research-consented, de-identified organ transplant donors through a collaboration with the UCSF VITAL Core (https://surgeryresearch.ucsf.edu/laboratories-research-centers/vital-core.aspx) and designated as non-human subjects research specimens by the UCSF IRB.

All experiments with animals were approved by the Institutional Animal Care and Use Committees at the University of California, San Francisco (AN183237-03) and the University of Calgary (AC16-0204).

## Decision letter and Author response

Decision letter https://doi.org/10.7554/eLife.78915.sa1
Author response https://doi.org/10.7554/eLife.78915.sa2

## Additional files

### Supplementary files
• Transparent reporting form
• MDAR checklist

### Data availability

Data generated or analyzed during this study are included in the manuscript. Previously published RNAseq datasets were deposited in GEO under accession codes GSE72166 and GSE76579.

The following previously published datasets were used:

| Author(s) | Year | Dataset title | Dataset URL | Database and Identifier |
|---|---|---|---|---|
| Chang AJ, Ortega FE, Riegler J, Madison DV, Krasnow MA | 2015 | Expression profile of mouse carotid body and adrenal medulla | https://www.ncbi.nlm.nih.gov/geo/query/acc.cgi?acc=GSE72166 | NCBI Gene Expression Omnibus, GSE72166 |
| Matsunami H, Zhou T, Chien M | 2016 | Single cell transcriptome analysis of mouse carotid body glomus cells | https://www.ncbi.nlm.nih.gov/geo/query/acc.cgi?acc=GSE76579 | NCBI Gene Expression Omnibus, GSE76579 |

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
