## [Editor Report]

The arterial chemoreceptors are the body's primary defense against hypoxia. In particular, the carotid body glomus cells (type 1) are highly sensitive to decreases in oxygen availability where the mitochondria and plasma membrane signaling proteins have been implicated in oxygen sensing by type 1 cells. Here, Chang and colleagues identified HIGD1C, a novel hypoxia-inducible gene domain factor isoform, is essential for carotid body oxygen sensing, where it enhances complex IV sensitivity to hypoxia. Discovery of this protein and its function brings back into focus the importance of how specialized mitochondria can act as sensors to metabolic stresses like hypoxia.

---

## [Decision Letter]

**Decision letter after peer review:**

Thank you for sending your article entitled "Tissue-specific mitochondrial HIGD1C promotes oxygen sensitivity in carotid body chemoreceptors" for peer review at *eLife*. Your article is being evaluated by 3 peer reviewers, one of whom is a member of our Board of Reviewing Editors, and the evaluation is being overseen by a Reviewing Editor and Vivek Malhotra as the Senior Editor.

I am detailing below the major revisions requested by the reviewers. The list appears to be long, but many of the concerns can be addressed by rewriting and clarifying the text. The full comments of the reviewers follow the list of major revisions requested.

Major concerns

1. Parts of the manuscript are difficult to follow re data presentation since the data are densely reported in places. Eg Figure 1 Figure supplement 2 on page 9 is virtually impossible to read.

2. A tighter introduction is required re references where credit should be given to the early studies showing that hypoxia excites CB and breathing and chemodenervation abolishes the breathing responses (eg Neil; Torrance; Lahiri should be given credit in para 1 on page 3).

3. Additionally, primary references should be given for statements of fact like …'Under these pathological conditions, suppressing CB activity improves causal symptoms such as hypertension (REF), cardiac arrhythmias (REF), and insulin resistance (REF) (Iturriaga, 2018)'. Credit must be given to the primary source, not a review.

4. p 4 'We found that three such genes, Higd1c, Cox4i2, and Ndufa4l2, were expressed at higher levels in the mouse CB (Figure 1A)'. Are you confident that this expression is only in type 1 cells and not type 2 cells?

5. Much of the data presented has used parametric statistics on n of 3/3 or 4/4 samples eg Figure 1 supp 3 page 10. Are these data normally distributed to support parametric use of stats? Please comment on the sample re biological replicates v technical replicates. How were the data treated here?

6. The gel on p26 panel B looks over run. Comment?

7. Aspects of the data are very convincing. Whilst this large body of work shows the putative role of HIGD1C in oxygen sensing, the relationship between this protein and the graded response of afferent nerve firing to physiological levels of hypoxia remains to be established to support an essential role for this signalling pathway in hypoxic chemotransduction. This limitation should be acknowledged. Please comment?

8. With respect to the effects of Higd1c KO on ventilation. Although all KO's appear to have a blunted ventilatory response to hypoxia, a ventilatory response remains none the less. This is particularly evident for the 1.1 KO as shown in the example trace in Figure 2 supplement 1 A where the second response to a hypoxic stimulus looks very similar between the KO and normal mouse. This is a somewhat disappointing result in that it may limit the conclusions that can be drawn from this study. One could have hoped that a Higd1c KO would either eliminate the response to hypoxia or had no effect at all. As this is a very important piece of evidence I want to ask for more detail regarding the data and methods.

9. Forgive me if I have misunderstood but in the methods section you state that "we used a respiratory rate of 215 breaths/min (comparable to the mean of a wild type in hypoxia) as a cut off for inclusion in this study". It is unclear how this works. Are you rejecting recordings with RR in excess of this figure or below this figure? When are you making this decision, under normoxic conditions at the beginning of an experiment, under hypoxic conditions, or at any time in the recording? Can you clarify this point and indicate how often recordings were rejected by this criteria in the different mice?

10. Was there any notable difference in mouse behaviour between genotypes during plethysmography? Mice usually become much less active in hypoxia and basal metabolic rate goes down resulting in a fall in CO2 production and a drop in PaCO2 which leads to a fall in ventilation (hypoxic ventilatory decline: HVD). This often complicates the analysis/interpretation of responses to hypoxia. Over what time period do you measure changes in ventilation – over the entire period of exposure to hypoxia, at the peak? Or towards the end? Are changes in the rate of HVD influencing the comparison between KO and wt? Have you measured PaCO2 under hypoxic conditions? I would predict that it would be lower than under control conditions. Could you introduce a little CO2 in hypoxic conditions to try to neutralise this effect?

11. What happens if you do a sham switch (i.e. switch between two identical gasses/air), is there any startle response associated with just mechanically switching between two gas lines? If not what happens if you look specifically at the peak/early response to hypoxia before CO2 has much chance to fall?

12. Are Higd1c +/+ mice taken from your in house breeding programs generating Higd1c -/- mice or are they just off the shelf C57Bl/6J from Jax? If the latter how long are these mice held in your facility (under the same conditions as the KO mice) before being used?

13. In the methods section on nerve recordings you state that "Preparations were exposed to a brief hypoxic challenge (PO2 = 60 mmHg) to determine viability; preparations that failed to show a clear increase in activity during this challenge were discarded". You are testing the hypothesis that Higd1c is important/essential for the generation of the response to hypoxia so why exclude data from preparations that lack a hypoxic response? This is just what you are looking for! I hope this is just a mistake in writing up the methods, BUT if this is really what was done then the whole nerve recording study could be invalid (depending on just what proportion of preparations were actually discarded). If you want some sort of test for viability of the preparation, which is a good idea, then there are many other stimuli one could choose instead that do not involve mitochondrial function e.g hypercapnia/acidosis, high potassium, acetyl choline, TASK channel inhibitors/respiratory stimulants.

14. One of the key issues regarding the role of alternative mitochondrial subunits in the carotid body is not simply that they may affect the maximal turnover rate of cytochrome oxidase but whether they alter the kinetic parameters defining sensitivity to hypoxia (i.e. Km, oxygen affinity, P50 ). To determine these really requires making a number of measurements at different PO2 including a zero point (anoxia). This could have been done for both nerve recordings and calcium measurements. Unfortunately it was not, so we are left with data from both of these experiments which demonstrates that the response to hypoxia is smaller in the KO but we have no idea whether these kinetic parameters have changed or not. The study would be a lot stronger if you could demonstrate a change in Km in the Higd1c KO.

15. It would appear from the results obtained with Rh123 and the weak responses to FCCP that there was inadequate loading of this dye to make good measurements of mitochondrial potential (ψM) in many of the cells studied. If you look at the paper by Biscoe and Duchen they were achieving a 100% increase in Rh125 fluorescence upon adding FCCP, Anoxia or CN, not 20%. The level of loading is critical with this dye in order to get it to work properly in the de-quench mode. In addition, to correct for differences in dye loading, responses to acute hypoxia should probably be normalised not only to the baseline but also to the response to FCCP. The smaller response to hypoxia in Higd1c -/- could simply reflect lower dye loading, with consequent lower dynamic response from Rh123 to change in mitochondrial potential, rather than any actual change in ψM.

16. The response to anoxia has not been tested which again confounds attempts to determine a P50 for mitochondrial depolarisation. I note that the rate of change in PO2 (Figure 4 suppl 1 A) is rather slow so some Rh123 may also leak out of the cell during the recording. Were any corrections applied for dye leakage?

17. In summary the interpretation of this data is equivocal. Higd1c -/- may take up Rh123 less avidly than wt but is this due to a lower resting ψM or something else? This experiment needs repeating extending loading time or Rh123 concentration until robust responses to FCCP can be recorded.

18. If you want to see if genotype affects resting ψM you would probably be better not working in the quench mode but either using much lower concentrations of Rh123 or another probe, e.g. TMRM, and loading for a longer period of time (until dye uptake reaches a steady state without any quenching occuring).

19. Data in Fig6-E shows >= 40% inhibition of oxygen consumption under hypoxic conditions compared to normoxia in all HEK cell types studied. Given that the level of oxygen in hypoxia is stated to be 25 mmHg this is something of a surprise. The P50 for most cells is thought to be < 1mmHg. Something is wrong here. If you are using an Oroboros Oxygraph (as stated in methods) it should be possible to measure oxygen consumption from air all the way down to zero oxygen and derive an exact P50 for each cell type. This should be done. This is a very important experiment that proports to show that it is the combination of Higd1c and Cox4I2 that generates the unusual sensitivity of complex IV towards oxygen. Philosophically I like this hypothesis. If true it would mark a major advance in this field, but I want to see some hard data with rigorously determined kinetic measurements!

20. It would be better in future if the calcium imaging studies could be performed using the same perfused preparation that the nerve recording experiments used. This would remove the requirement for equilibrating superfusate with 95% oxygen which is far from physiological and would allow the carotid body tissue to be perfused with physiologically relevant levels of oxygen. However, superfusion with hyperoxic solutions is considered standard in many systems and so additional experiments are not required at this time.

21. Expanding the study to see if HIGD1A is expressed in carotid bodies of multiple species would strengthen the paper. It would also be interesting to see if it is absent in the carotid bodies of guinea pigs which are not acutely oxygen sensitive. Furthermore, chromaffin cells in fetal adrenal medulla are oxygen-sensitive whereas mature chromaffin cells are not, it would be fascinating to see if HIGD1A expression changes during the maturation of chromaffin cells. These possibilities might be discussed.

22. Tidal volume changes with the weight of the animals. Tidal volume should be adjusted for the weights of the animals tested. If there are weight differences between knockouts and wildtype it can have profound effects on the data.

*Reviewer #1 (Recommendations for the authors):*

Understanding the precise intracellular signalling pathways underpinning hypoxic sensitivity in the carotid body chemoreceptor has been a major unsolved area in sensory neurophysiology. Early studies almost 60 years ago suggested the importance of a metabolic signal coupled to the mitchrondria, which could regulate intracellular calcium homeostasis and exocystosis of excitatory transmitter in type I glomus cells.

Using RNAseq this manuscript has identified a novel gene that encodes protein regulation of Complex IV in the carotid body response to hypoxia. Here the authors found HIGD1C, a novel hypoxia-inducible gene domain factor isoform, as an electron transport chain Complex IV-interacting protein was almost exclusively expressed in the carotid body. Using a combination molecular biology, protein chemistry, genetic knock-out, calcium imaging and respiratory measurements, the authors elegantly demonstrated the physiological utlility of HIGD1C being required for carotid body oxygen sensing via enhanced Complex IV sensitivity to hypoxia. Deletion of the gene massively attenuated the chemoreceptor response to hypoxia, whereas of expression of HIGD1C could recapitulate the increased oxygen sensitivity of Complex IV in HEK 293T cells, but not the response to hypercapnia.

Whilst this large body of work shows the putative role of HIGD1C in oxygen sensing, the relationship between this protein and the graded response of afferent nerve firing to physiological levels of hypoxia remains to be established to support an essential role for this signalling pathway in hypoxic chemotransduction.

This is a large body of work supporting the idea that HIGD1C is required for hypoxic sensing in the mouse through the regulation complex IV. The experiments are well designed, complex and appear to have been carefully performed.

Specific comments for improvement:

Parts of the manuscript are difficult to follow re data presentation since the data are densely reported in places. Eg Figure 1 Figure supplement 2 on page 9 is virtually impossible to read.

A tighter introduction is required re references where credit should be given to the early studies showing that hypoxia excites CB and breathing and chemodenervation abolishes the breathing responses (eg Neil; Torrance; Lahiri should be given credit in para 1 on page 3).

Additionally, primary references should be given for statements of fact like …'Under these pathological conditions, suppressing CB activity improves causal symptoms such as hypertension (REF), cardiac arrhythmias (REF), and insulin resistance (REF) (Iturriaga, 2018)'. Credit must be given to the primary source, not a review.

p 4 'We found that three such genes, Higd1c, Cox4i2, and Ndufa4l2, were expressed at higher levels in the mouse CB (Figure 1A)'. Are you confident that this expression is only in type 1 cells and not type 2 cells?

Much of the data presented has used parametric statistics on n of 3/3 or 4/4 samples eg Figure 1 supp 3 page 10. Are these data normally distributed to support parametric use of stats? Please comment on the sample re biological replicates v technical replicates. How were the data treated here?

The gel on p26 panel B looks over run. Comment?

Aspects of the data are very convincing. Whilst this large body of work shows the putative role of HIGD1C in oxygen sensing, the relationship between this protein and the graded response of afferent nerve firing to physiological levels of hypoxia remains to be established to support an essential role for this signalling pathway in hypoxic chemotransduction. This limitation should be acknowledged. Please comment?

*Reviewer #2 (Recommendations for the authors):*

I have a number of major concerns about parts of this research which need to be addressed before I could comment on its conclusions and importance. Some of these may just require a better explanation or correction to the text (e.g. where there may be an ambiguity over how an experiment was conducted). Others may require further work

Major comments/Questions

General reporting of results

A very general comment I have about this paper is that numerous experiments, particularly in the latter half of this paper using HEK cells, have been conducted with an 'n' of only 3. I do not know what *eLife*s's normal expectations/requirements are but I would have thought that an 'n' of at least 4 or 5 independent observations should normally be required. Similarly, numerous gels/blots are presented without any indication of how often these types of experiments were repeated. This should be reported and the journal should have some policy over what is expected. To my mind there could be a substantial amount of further work that needs to be completed before this study could be formally published. Apart from this issue the statistics seem to have been done with suitable care and rigour.

Questions/suggested changes.

Plethysmography.

With respect to the effects of Higd1c KO on ventilation. Although all KO's appear to have a blunted ventilatory response to hypoxia, a ventilatory response remains none the less. This is particularly evident for the 1.1 KO as shown in the example trace in Figure 2 supplement 1 A where the second response to a hypoxic stimulus looks very similar between the KO and normal mouse. This is a somewhat disappointing result in that it may limit the conclusions that can be drawn from this study. One could have hoped that a Higd1c KO would either eliminate the response to hypoxia or had no effect at all. As this is a very important piece of evidence I want to ask for more detail regarding the data and methods.

Forgive me if I have misunderstood but in the methods section you state that "we used a respiratory rate of 215 breaths/min (comparable to the mean of a wild type in hypoxia) as a cut off for inclusion in this study". It is unclear how this works. Are you rejecting recordings with RR in excess of this figure or below this figure? When are you making this decision, under normoxic conditions at the beginning of an experiment, under hypoxic conditions, or at any time in the recording? Can you clarify this point and indicate how often recordings were rejected by this criteria in the different mice?

Was there any notable difference in mouse behaviour between genotypes during plethysmography? Mice usually become much less active in hypoxia and basal metabolic rate goes down resulting in a fall in CO2 production and a drop in PaCO2 which leads to a fall in ventilation (hypoxic ventilatory decline: HVD). This often complicates the analysis/interpretation of responses to hypoxia. Over what time period do you measure changes in ventilation – over the entire period of exposure to hypoxia, at the peak? Or towards the end? Are changes in the rate of HVD influencing the comparison between KO and wt? Have you measured PaCO2 under hypoxic conditions? I would predict that it would be lower than under control conditions. Could you introduce a little CO2 in hypoxic conditions to try to neutralise this effect?

What happens if you do a sham switch (i.e. switch between two identical gasses/air), is there any startle response associated with just mechanically switching between two gas lines? If not what happens if you look specifically at the peak/early response to hypoxia before CO2 has much chance to fall?

Are Higd1c +/+ mice taken from your in house breeding programs generating Higd1c -/- mice or are they just off the shelf C57Bl/6J from Jax? If the latter how long are these mice held in your facility (under the same conditions as the KO mice) before being used?

Nerve fibre recordings and calcium recordings.

In the methods section on nerve recordings you state that "Preparations were exposed to a brief hypoxic challenge (PO2 = 60 mmHg) to determine viability; preparations that failed to show a clear increase in activity during this challenge were discarded". You are testing the hypothesis that Higd1c is important/essential for the generation of the response to hypoxia so why exclude data from preparations that lack a hypoxic response? This is just what you are looking for! I hope this is just a mistake in writing up the methods, BUT if this is really what was done then the whole nerve recording study could be invalid (depending on just what proportion of preparations were actually discarded). If you want some sort of test for viability of the preparation, which is a good idea, then there are many other stimuli one could choose instead that do not involve mitochondrial function e.g hypercapnia/acidosis, high potassium, acetyl choline, TASK channel inhibitors/respiratory stimulants.

One of the key issues regarding the role of alternative mitochondrial subunits in the carotid body is not simply that they may affect the maximal turnover rate of cytochrome oxidase but whether they alter the kinetic parameters defining sensitivity to hypoxia (i.e. Km, oxygen affinity, P50 ). To determine these really requires making a number of measurements at different PO2 including a zero point (anoxia). This could have been done for both nerve recordings and calcium measurements. Unfortunately it was not, so we are left with data from both of these experiments which demonstrates that the response to hypoxia is smaller in the KO but we have no idea whether these kinetic parameters have changed or not. The study would be a lot stronger if you could demonstrate a change in Km in the Higd1c KO.

Rh123 experiments.

It would appear from the results obtained with Rh123 and the weak responses to FCCP that there was inadequate loading of this dye to make good measurements of mitochondrial potential (ψM) in many of the cells studied. If you look at the paper by Biscoe and Duchen they were achieving a 100% increase in Rh125 fluorescence upon adding FCCP, Anoxia or CN, not 20%. The level of loading is critical with this dye in order to get it to work properly in the de-quench mode. In addition, to correct for differences in dye loading, responses to acute hypoxia should probably be normalised not only to the baseline but also to the response to FCCP. The smaller response to hypoxia in Higd1c -/- could simply reflect lower dye loading, with consequent lower dynamic response from Rh123 to change in mitochondrial potential, rather than any actual change in ψM.

The response to anoxia has not been tested which again confounds attempts to determine a P50 for mitochondrial depolarisation. I note that the rate of change in PO2 (Figure 4 suppl 1 A) is rather slow so some Rh123 may also leak out of the cell during the recording. Were any corrections applied for dye leakage?

In summary the interpretation of this data is equivocal. Higd1c -/- may take up Rh123 less avidly than wt but is this due to a lower resting ψM or something else? This experiment needs repeating extending loading time or Rh123 concentration until robust responses to FCCP can be recorded.

If you want to see if genotype affects resting ψM you would probably be better not working in the quench mode but either using much lower concentrations of Rh123 or another probe, e.g. TMRM, and loading for a longer period of time (until dye uptake reaches a steady state without any quenching occuring).

CIV sensitivity to Hypoxia in HEK.

Data in Fig6-E shows >= 40% inhibition of oxygen consumption under hypoxic conditions compared to normoxia in all HEK cell types studied. Given that the level of oxygen in hypoxia is stated to be 25 mmHg this is something of a surprise. The P50 for most cells is thought to be < 1mmHg. Something is wrong here. If you are using an Oroboros Oxygraph (as stated in methods) it should be possible to measure oxygen consumption from air all the way down to zero oxygen and derive an exact P50 for each cell type. This should be done. This is a very important experiment that proports to show that it is the combination of Higd1c and Cox4I2 that generates the unusual sensitivity of complex IV towards oxygen. Philosophically I like this hypothesis. If true it would mark a major advance in this field, but I want to see some hard data with rigorously determined kinetic measurements!

*Reviewer #3 (Recommendations for the authors):*

This paper uses a range of techniques to demonstrate the presence of a novel protein (HIGD1C) that interacts with complex 4 of the mitochondrial electron transport chain. The authors demonstrate that this protein is required for hypoxic chemotransduction at the level of the whole animal (plethysmography), at the level of the in-vitro organ (carotid sinus nerve recordings) and at the level of the glomus cell (calcium imaging). The authors then go on to demonstrate that HIGD1C may interact and alter the sensitivity of complex 4 to hypoxia.

The authors are to be congratulated on a set of thorough physiological experiments that are then extended by detailed cellular respiration studies. Working with mouse carotid body tissue is incredibly challenging and the authors have done extremely well to get such high quality and valuable data.

The combination of genetic, physiological and cellular experiments make this an extremely compelling paper suitable for publication in *eLife*.

The question of why carotid body glomus cells are unusually sensitive to hypoxia has troubled researchers for decades. This paper makes an extremely compelling case for the expression of HIGD1C modulating complex 4 and thereby sensitizing it to hypoxia. The data seems strong and the paper is likely to have a major impact in the field of oxygen sensing.

There are several observations that would greatly strengthen the paper.

1. It would be better in future if the calcium imaging studies could be performed using the same perfused preparation that the nerve recording experiments used. This would remove the requirement for equilibrating superfusate with 95% oxygen which is far from physiological and would allow the carotid body tissue to be perfused with physiologically relevant levels of oxygen. However, superfusion with hyperoxic solutions is considered standard in many systems and so additional experiments are not required at this time.

2. Expanding the study to see if HIGD1C is expressed in carotid bodies of multiple species would strengthen the paper. It would also be interesting to see if it is absent in the carotid bodies of guinea pigs which are not acutely oxygen sensitive. Furthermore, chromaffin cells in fetal adrenal medulla are oxygen-sensitive whereas mature chromaffin cells are not, it would be fascinating to see if HIGD1C expression changes during the maturation of chromaffin cells. These possibilities might be discussed.

3. Tidal volume changes with the weight of the animals. Tidal volume should be adjusted for the weights of the animals tested. If there are weight differences between knockouts and wildtype it can have profound effects on the data.

---

## [Author Response]

Major concerns1. Parts of the manuscript are difficult to follow re data presentation since the data are densely reported in places. Eg Figure 1 Figure supplement 2 on page 9 is virtually impossible to read.

We strove to make the figures more legible in this revision. For example, we enlarged and redistributed the panels in Figure 1-figure supplement 2. We split the Figure 1-figure supplement 3 into two Figure supplements (Figure 1-figure supplements 3 and 5). In addition, we have now submitted full-page versions of figures that can be resized as needed.

2. A tighter introduction is required re references where credit should be given to the early studies showing that hypoxia excites CB and breathing and chemodenervation abolishes the breathing responses (eg Neil; Torrance; Lahiri should be given credit in para 1 on page 3).

We now reference the following primary papers for key early observations showing that carotid body stimulated afferent nerve activity and ventilation that can be abolished by chemo- denervation or carotid body ablation (p. 3):

De Castro F. (1928). Sur la structure et l'innervation du sinus carotidien de l'homme et des mammifères. Nouveaux faits sur l'innervation et la fonction du glomus caroticum. Trav. Lab. Rech. Biol. 25, 331–380

Heymans C., Bouckaert J. J., Dautrebande L. (1931). Au sujet du mecanisme de la bradycardie provoquée par la nicotine, la lobéline, le cyanure, le sulfure de sodium, les nitrites et la morphine, et de la bradycardie asphyxique. *Arch. Int. Pharmacodyn.* 41, 261–289

Neil and O’Regan, J Physiol, 1971, 215: 33-47

Black, McCloskey, and Torrance, Respir Physiol, 1971, 13: 36-49

Lahiri and DeLaney, Respir Physiol, 1975, 24: 249-266

Lahiri and DeLaney, Respir Physiol, 1975, 24:2 67-286

Verna, Roumy, and Leitner, Brain Res, 1975, 100: 13-23

We have also added this review of the history:

De Castro, F. (2009). The discovery of sensory nature of the carotid bodies--invited article. *Adv Exp Med Biol, 648*, 1-18

3. Additionally, primary references should be given for statements of fact like …'Under these pathological conditions, suppressing CB activity improves causal symptoms such as hypertension (REF), cardiac arrhythmias (REF), and insulin resistance (REF) (Iturriaga, 2018)'. Credit must be given to the primary source, not a review.

We now reference the primary sources for these observations (p. 3):

Hypertension

Fletcher et al., J Appl Physiol, 1992

Del Rio et al., Hypertension, 2016

Abdala et al., J Physiol, 2012

Narkiewiecz et al., 2016, JACC Basic Transl Sci

Cardiac arrhythmias

Marcus et al., 2014, J Physiol

Del Rio et al., 2013, J Am Coll Cardiol

Insulin resistance

Ribeiro et al., 2013, Diabetes

Sacramento et al., 2017, Diabetelogica

4. p 4 'We found that three such genes, Higd1c, Cox4i2, and Ndufa4l2, were expressed at higher levels in the mouse CB (Figure 1A)'. Are you confident that this expression is only in type 1 cells and not type 2 cells?

RNAseq of whole carotid body tissue (Figure 1A) cannot determine if the expression of these genes is restricted to any particular cell type. To overcome this, we performed duplex in situ hybridizations to show that *Higd1c* mRNA co-localizes with *Th* mRNA in type 1 cells (Figure 1D, E). Additional support comes from single-cell RNA studies that show high levels of expression of all three genes in glomus cells (Zhou et al., 2016, J Physiol). The pattern of gene expression of *Cox4i2* and *Ndufa4l2* we detected by in situ hybridization and immunostaining are consistent with the expression of these genes in type 1 cells (Zhou et al., 2016, J Physiol; Moreno-Dominguez et al., 2020, Sci Signal). To definitively show expression of *Higd1c* in only type 1 cells and not in both type 1 and type 2 cells in tissue sections is not possible because there is no antibody available for HIGD1C. We have added this limitation in the Discussion (p. 41) and raised the possibility that *Higd1c* could be expressed in both type 1 and type 2 cells, as both cell types have been proposed to work together in sensory signaling in hypoxia (Leonard et al., 2018, Front Physiol).

5. Much of the data presented has used parametric statistics on n of 3/3 or 4/4 samples eg Figure 1 supp 3 page 10. Are these data normally distributed to support parametric use of stats? Please comment on the sample re biological replicates v technical replicates. How were the data treated here?

All samples in the experiments presented were biological replicates, and all groups were analyzed by the Shapiro-Wilk test to check for normality before running parametric or non-parametric tests as appropriate (described in p. 59). We have increased the n (number of samples) to 5 or 6 samples per tissue for the RT-qPCR experiments in Figure 1-figure supplement 3 and checked for normality to justify the use of parametric tests.

6. The gel on p26 panel B looks over run. Comment?

This appearance is due to the large amount of mitochondria required to detect the mouse version of HIGD1C. This could be due to a reduced ability of mouse HIGD1C to assemble with human Complex IV proteins expressed endogenously by HEK293T cells.

7. Aspects of the data are very convincing. Whilst this large body of work shows the putative role of HIGD1C in oxygen sensing, the relationship between this protein and the graded response of afferent nerve firing to physiological levels of hypoxia remains to be established to support an essential role for this signalling pathway in hypoxic chemotransduction. This limitation should be acknowledged. Please comment?

We greatly appreciate the reviewer’s assessment of the quality of our data. We showed defects in the graded response of the carotid sinus nerve in *Higd1c* mutants to three physiological levels of hypoxia (PO_2_=80, 60, and 40 mmHg) in Figure 3A-D that stimulate carotid body sensory output and ventilation.

8. With respect to the effects of Higd1c KO on ventilation. Although all KO's appear to have a blunted ventilatory response to hypoxia, a ventilatory response remains none the less. This is particularly evident for the 1.1 KO as shown in the example trace in Figure 2 supplement 1 A where the second response to a hypoxic stimulus looks very similar between the KO and normal mouse. This is a somewhat disappointing result in that it may limit the conclusions that can be drawn from this study. One could have hoped that a Higd1c KO would either eliminate the response to hypoxia or had no effect at all. As this is a very important piece of evidence I want to ask for more detail regarding the data and methods.

Over the last several years, a nuanced view of oxygen sensing has emerged in which there are likely multiple sites beyond the carotid body and multiple molecules (even within the carotid body) involved in acute oxygen sensitivity (reviewed in Funk and Gourine, 2017, J Appl Physiol; Holmes et al., 2022, Front Physiol; Barioni et al., 2022, Sci Adv). The carotid body glomus cell mitochondrial electron transport chain is expected to be the most sensitive site, and we propose that HIGD1C is present almost exclusively in the carotid bodies when acting in combination with less exclusively (but by no means broadly) expressed ETC proteins such as COX4I2 and NDUFA4L2, is responsible for this heightened sensitivity.

With regards to multiple sites, while the carotid bodies are the major oxygen and likely the most sensitive chemoreceptors contributing to the HVR, we do not exclude a role for other oxygen-sensitive sites such as the aortic bodies, brainstem, and spinal cord (reviewed in Funk and Gourine, 2017, J Appl Physiol; Hodges and Forster, 2012, Neur Regen Res). Indeed, our recent work describes spinal oxygen sensors that can stimulate respiratory centers in the brainstem (Barioni et al., 2022, Sci Adv).

With regards to multiple molecules involved in acute oxygen sensitivity, please note Figure 3-figure supplement 1. In glomus cells from *Higd1c^+/+^* animals, a plethora of cells respond strongly to both severe (PO_2_=25 mmHg) and less severe (PO_2_=50 mmHg) hypoxia, whereas cells from *Higd1c^-/-^* mostly respond (though be it at reduced intensity) to only severe hypoxia. This parallels the data from carotid sinus nerve recordings in Figure 3A-D. These data suggest that other molecules are capable of partially subserving HIGD1C’s role in severe hypoxia.

With regards to the investigation of HVR in *Higd1c^-/-^* animals, the magnitude of the defect in the hypoxic ventilatory response we report was at least as severe as in other rodent studies in which carotid bodies were denervated or ablated (Soliz et al., 2005, J Physiol; Del Rio et al., 2013, J Am Coll Cardiol). Moreover, the literature suggests that all experimental animal species show some degree of recovery of the HVR over time after carotid body resection or denervation when tested in an awake, unanesthetized state. Therefore, we might expect some degree of compensation by other sites/molecules in the absence of carotid body HIGD1C.

For the time course data in Figure 2-figure supplement 1A-C, please note that the last (third) challenge in Figure 2-figure supplement 1A-C is hypercapnia. The most pertinent panel is C, which shows minute ventilation (volume exchanged per minute=respiratory rate*tidal volume). Animals can increase ventilation by increasing either the respiratory rate, tidal volume, or both, and minute ventilation accounts for both respiratory rate and tidal volume changes. The difference between minute ventilation of *Higd1c^+/+^* and *Higd1c^-/-^* animals was apparent and consistent throughout the entire duration of hypoxic periods, including ramp periods from normoxia (21% O_2_) to hypoxia (10% O_2_). See Figure 2-figure supplement 1A-C with 30 s bins for normoxic (Nox, yellow) and hypoxic (Hox, blue) periods of the time course. Ramps (no color) were 1 min. Data are shown as mean and SEM.

We have replaced Figure 2-figure supplement 1A-C because they better illustrate the reduction in HVR.

9. Forgive me if I have misunderstood but in the methods section you state that "we used a respiratory rate of 215 breaths/min (comparable to the mean of a wild type in hypoxia) as a cut off for inclusion in this study". It is unclear how this works. Are you rejecting recordings with RR in excess of this figure or below this figure? When are you making this decision, under normoxic conditions at the beginning of an experiment, under hypoxic conditions, or at any time in the recording? Can you clarify this point and indicate how often recordings were rejected by this criteria in the different mice?

We rejected trials where any of the normoxic periods had an average respiratory rate that exceeded 215 breath/min in order to assess calm breathing and exclude artefacts from sniffing, grooming, and movement that correlated with high-frequency events above the ventilation in hypoxia. Normoxic periods evaluated included the baseline at the beginning of the experiment as well as subsequent inter-stimulus normoxic periods. Ventilation in hypoxia was much more regular with few artefacts, and high frequency in hypoxia was not used as an exclusion criterion. We have now stated this exclusion criterion and our rationale more clearly in the Methods (p. 47).

Author response image 1 shows the percentage of trials rejected by genotype (number of trials indicated on bars). The percent rejected between *Higd1c^+/+^* and *Higd1c^-/-^* animals by allele are not statistically significant by the *z* test of proportions (p=0.7039, 0.5353, and 0.7114 for *1-1*, *3-1*, and *5-3*, respectively). We have added these stats to the Methods (p. 47).

**Author response image 1. sa2fig1:** 

10. Was there any notable difference in mouse behaviour between genotypes during plethysmography? Mice usually become much less active in hypoxia and basal metabolic rate goes down resulting in a fall in CO2 production and a drop in PaCO2 which leads to a fall in ventilation (hypoxic ventilatory decline: HVD). This often complicates the analysis/interpretation of responses to hypoxia. Over what time period do you measure changes in ventilation – over the entire period of exposure to hypoxia, at the peak? Or towards the end? Are changes in the rate of HVD influencing the comparison between KO and wt? Have you measured PaCO2 under hypoxic conditions? I would predict that it would be lower than under control conditions. Could you introduce a little CO2 in hypoxic conditions to try to neutralise this effect?

No behavioral differences between *Higd1c^+/+^* and *Higd1c^-/-^* animals during our whole-body plethysmography experiments were observed. Both genotypes became less active in hypoxia and exhibited more regular breathing.

To calculate the ventilation data for hypoxia, we first calculated the mean over the entire duration of both hypoxic stimulus periods (5 min each), and then we averaged the mean values of the two hypoxic periods. The 1-min ramp periods for the chamber to go from normoxia (21% O_2_) to hypoxia (10% O_2_) or vice versa were not included in calculations for either normoxia or hypoxia (see also time courses in response to item 8). From the time course data (see item 8), *Higd1c^-/-^* mutants had lower ventilation at all time points in hypoxia and the ramp to hypoxia.

We agree with the reviewer that hypoxia will reduce PaCO_2_. On the one hand, hypoxia acts to attenuate ventilation by suppressing metabolism (reducing CO_2_ production) and directly inhibiting neurons of the respiratory controller. On the other hand, this attenuation of ventilation is offset by central and peripheral oxygen chemoreceptors which excite the respiratory control system. In neonates, the oxygen chemoreceptors reduce but do not fully overcome the central suppressive effect of hypoxia; in adults, oxygen chemoreceptors are more potent and can fully overcome the hypoxic depression of neurons to cause a net increase in ventilation. Thus, in adults, hypoxia will cause a net reduction in PaCO_2_, a respiratory stimulant. As the HVR is lower in *Higd1c^-/-^* animals, we would expect *Higd1c^-/-^* to have a higher PaCO_2_ in hypoxia than *Higd1c^+/+^*_,_ providing more respiratory stimulant to *Higd1c^-/-^* animals. Thus, the lower HVR during poikilocapnic hypoxia conditions in *Higd1c^-/-^* compared to *Higd1c^+/+^* is expected to *underestimate* the importance of HIGD1C to the hypoxic response at the level of the oxygen chemoreceptor.

We have not attempted to regulate PaCO_2_ under hypoxic conditions in our mice. In large animals, blood gases can be monitored in real-time, allowing the use of feedback control to deliver appropriate CO_2_ to maintain PaCO_2_ at a constant level. This is not feasible in mice. Blood gas measurements in mice are terminal experiments that require many animals to get a single time point, more than we can breed and raise to adults in two months. In addition, collecting sufficient arterial blood for blood gas experiments in adult mice requires anesthesia, which alters the HVR, or installing carotid artery catheters in awake mice, which would ablate one carotid body.

Introducing a fixed level of CO_2_ during hypoxia is sometimes performed to try to mitigate the reduction in PaCO_2_ due to hyperventilation. However, such a maneuver is expected to (a) stimulate the carotid body in an oxygen chemoreceptor-dependent manner because of the multiplicative O_2_ and CO_2_ interaction at the carotid body, the mechanism for which is unknown, and (b) change the non-additive (likely hypoadditive) interaction between central and peripheral chemoreceptors which has not been carefully characterized in mice. Together, these effects are almost impossible to quantify. Rather than adding unknowns, we chose to opt for simplicity and perform our experiments under poikilocapnic hypoxia conditions that mimic high altitude (i.e., hypoxia without CO_2_ supplementation). At high altitudes, carotid body oxygen sensing causes hyperventilation, and under these conditions, reducing oxygen sensing is expected to reduce ventilation.

11. What happens if you do a sham switch (i.e. switch between two identical gasses/air), is there any startle response associated with just mechanically switching between two gas lines? If not what happens if you look specifically at the peak/early response to hypoxia before CO2 has much chance to fall?

We did not observe any startle response associated with just switching between two identical gases. From the time course data (please see item 8), *Higd1c^-/-^* mutants exhibited lower ventilation at all time points, including during the ramp and first points of hypoxia.

12. Are Higd1c +/+ mice taken from your in house breeding programs generating Higd1c -/- mice or are they just off the shelf C57Bl/6J from Jax? If the latter how long are these mice held in your facility (under the same conditions as the KO mice) before being used?

*Higd1c^+/+^* and *Higd1c^-/-^* animals were generated from crosses between *Higd1c^+/-^* parents in our facility. We have added this detail to the Methods (p. 42).

13. In the methods section on nerve recordings you state that "Preparations were exposed to a brief hypoxic challenge (PO2 = 60 mmHg) to determine viability; preparations that failed to show a clear increase in activity during this challenge were discarded". You are testing the hypothesis that Higd1c is important/essential for the generation of the response to hypoxia so why exclude data from preparations that lack a hypoxic response? This is just what you are looking for! I hope this is just a mistake in writing up the methods, BUT if this is really what was done then the whole nerve recording study could be invalid (depending on just what proportion of preparations were actually discarded). If you want some sort of test for viability of the preparation, which is a good idea, then there are many other stimuli one could choose instead that do not involve mitochondrial function e.g hypercapnia/acidosis, high potassium, acetyl choline, TASK channel inhibitors/respiratory stimulants.

We thank the Reviewer for pointing out this mistake. Dr. Roy, who performed these challenging experiments, typically gives rat preparations a PO_2_=60 mmHg hypoxic pre-challenge, but he no longer does this in mice because the preparation is too delicate. We have removed this statement in the Methods and described the current practice (p. 49): “Recording was only attempted in nerves that survived cleaning and desheathing. The presence of action potentials under baseline conditions was used as the only test of preparation viability; data were obtained from all preparations deemed viable according to this criterion.”

We have also updated the representative traces in Figure 3A and B with graphs that better represents the mean data.

14. One of the key issues regarding the role of alternative mitochondrial subunits in the carotid body is not simply that they may affect the maximal turnover rate of cytochrome oxidase but whether they alter the kinetic parameters defining sensitivity to hypoxia (i.e. Km, oxygen affinity, P50 ). To determine these really requires making a number of measurements at different PO2 including a zero point (anoxia). This could have been done for both nerve recordings and calcium measurements. Unfortunately it was not, so we are left with data from both of these experiments which demonstrates that the response to hypoxia is smaller in the KO but we have no idea whether these kinetic parameters have changed or not. The study would be a lot stronger if you could demonstrate a change in Km in the Higd1c KO.

The main conclusion drawn from the calcium imaging and nerve recording experiments is that *Higd1c^-/-^* mutants are defective in graded responses to hypoxia in the physiological range (Figure 3). These experiments were performed in intact tissue to measure the activity of as many glomus cells as possible and to preserve normal cell-cell interactions that promote sensory signaling. However, because these experiments were performed in intact tissue, the PO_2_ experienced by individual glomus cells could not be accurately determined due to tissue heterogeneity. Thus, the PO_2_ of the perfusion reflected the upper limit of an unknown lower range of PO_2_ at the cellular level. Technically, anoxia is typically delivered in perfusion by adding a reducing agent like sodium dithionite to scavenge oxygen. However, such agents can also modify cell proteins and complicate the interpretation of results.

Furthermore, experiments at lower PO_2_ would not be informative for nerve recordings because carotid sinus nerve activity does not increase (and can even decrease) once PO_2_ is reduced below 30 mmHg (Fidone and Gonzalez, 1986, Handbook of Physiology, The Respiratory System; Hornbein et al., 1961, J Neurophysiol; Hornbein and Roos, 1963, J Physiol; Peng et al., 2020, J Neurophysiol). While intracellular calcium does continue to increase as PO_2_ is decreased below 30 mmHg, the lack of correlation of this additional intracellular calcium to sensory output would make it difficult to interpret these experiments (Peng et al., 2020, J Neurophysiol). We chose to focus on a more physiological PO_2_ range of 25-100 mmHg for carotid body sensory activity experiments, and in this range, *Higd1c^-/-^* mutants have reduced sensitivity in their response to hypoxia (Figure 3).

We thank the reviewer for the suggestion to curve fit our glomus cell imaging data to derive KmO_2_ values. Unfortunately, we do not have enough points in the near anoxic range to restrain a dose response curve for our current data. Dissociating large numbers of glomus cells to perform comprehensive studies at multiple oxygen levels is not feasible using adult mice, and thus, studies that report calcium and metabolic imaging from mouse glomus cells have not included this level of analysis (Fernandez-Augera et al., 2015, Cell Metab; Moreno-Dominguez et al., 2020, Sci Signal; Peng et al., 2020, J Neurophysiol, Peng et al., 2019, Respir Neurobiol Physiol; MacMillan et al., 2022, Commun Biol). The studies we have found that show derived KmO_2_ values utilized neonatal rats, which have larger carotid bodies and are easier to dissociate (Landuaer et al., 1995, J Physiol; Buckler and Turner, 2013, J Physiol).

We agree that determining whether alternative mitochondrial subunits change the Km of oxygen on the mitochondrial electron transport chain in glomus cells is an important question. However, the ideal experiment to make direct oxygen measurements on large numbers of purified glomus cells using cellular oxygen consumption to reduce oxygen levels is not currently possible in the mouse. We have added this limitation to the Discussion (p. 40): “Future studies of oxygen consumption by mitochondria of glomus cells, when feasible, will further illuminate the roles of these proteins in carotid body oxygen sensing.” Nevertheless, we have now determined the contribution of atypical mitochondrial proteins to Km of oxygen on electron transport chain activity in cell culture, where we could obtain more abundant material for direct oxygen measurements. We found that expression of both HIGD1C and COX4I2 in *HIGD1A*-KO HEK293T cells increased the p50_mito_ to enhance the sensitivity of the electron transport chain to hypoxia (see also response to item 19).

15. It would appear from the results obtained with Rh123 and the weak responses to FCCP that there was inadequate loading of this dye to make good measurements of mitochondrial potential (ψM) in many of the cells studied. If you look at the paper by Biscoe and Duchen they were achieving a 100% increase in Rh125 fluorescence upon adding FCCP, Anoxia or CN, not 20%. The level of loading is critical with this dye in order to get it to work properly in the de-quench mode.

It is difficult to directly compare our Rh123 experiments on intact mouse carotid bodies with experiments on dissociated rabbit glomus cells used in Biscoe and Duchen, 1992, J Physiol. We expect dye loading to be more heterogeneous in intact carotid bodies, compared to dissociated cells, due to the differential accessibility of different parts of the tissue to Rh123. We accounted for this heterogeneity by normalizing fluorescence to baseline. In addition, we do not know whether there are species-specific differences between mouse and rabbit or selection during dissociation and imaging for only a subset of glomus cells with better FCCP responses in Biscoe and Duchen. In developing our protocol, we tested multiple concentrations of Rh123 and incubation times, and increasing the time of dye loading did not increase baseline fluorescence, indicating we had reached a steady-state level of dye loading.

In addition, to correct for differences in dye loading, responses to acute hypoxia should probably be normalised not only to the baseline but also to the response to FCCP.

We are not confident that the FCCP response reflects a maximum mitochondrial membrane potential response against which the other responses could be normalized. FCCP was presented after both hypoxia and cyanide at the end of a long experiment, during which Rh123 fluorescence decreased over time.

The smaller response to hypoxia in Higd1c -/- could simply reflect lower dye loading, with consequent lower dynamic response from Rh123 to change in mitochondrial potential, rather than any actual change in ψM.

Because *Higd1c^+/+^* and *Higd1c^-/-^* carotid bodies were loaded with Rh123 under the same conditions and their morphologies and glomus cell counts were similar (Figure 1G-I), we do not believe that the smaller Rh123 responses of *Higd1c^-/-^* glomus cells to hypoxia was due to poorer dye loading but likely due to a combination of a less polarized inner mitochondrial membrane at baseline and weaker response to hypoxia. In *Higd1c^-/-^* glomus cells that had strong responses to FCCP comparable to *Higd1c^+/+^*, the response to hypoxia was still weaker than *Higd1c^+/+^* (Figure 4D-F). To resolve this, we performed additional experiments testing FCCP responses as the first stimulus in *Higd1c^+/+^* and *Higd1c^-/-^* carotid bodies and found that even under these more optimal conditions, there were smaller Rh123 responses to FCCP in *Higd1c^-/-^* glomus cells (p. 24, Figure 4-figure supplement 1D). This suggests that there is a defect in ψM in normoxia in *Higd1c^-/-^* glomus cells (see also response to item 17).

16. The response to anoxia has not been tested which again confounds attempts to determine a P50 for mitochondrial depolarisation. I note that the rate of change in PO2 (Figure 4 suppl 1 A) is rather slow so some Rh123 may also leak out of the cell during the recording. Were any corrections applied for dye leakage?

Yes, we did correct for dye leakage and bleaching by performing baseline subtraction with linear interpolation.

17. In summary the interpretation of this data is equivocal. Higd1c -/- may take up Rh123 less avidly than wt but is this due to a lower resting ψM or something else? This experiment needs repeating extending loading time or Rh123 concentration until robust responses to FCCP can be recorded.

We consider it likely that *Higd1c^-/-^* glomus cells had weaker responses to FCCP due to a less polarized inner mitochondrial membrane. HIGD1C could affect the function of the electron transport chain at all oxygen levels. To help resolve this, we performed additional experiments on *Higd1c^+/+^* and *Higd1c^-/-^* carotid bodies, testing FCCP responses as a first stimulus to estimate differences in resting membrane potential while minimizing dye leakage and bleaching over time. Similar to when FCCP was presented as the third stimulus (Figure 4C), *Higd1c^-/-^* glomus cells had smaller Rh123 responses to FCCP compared to *Higd1c^+/+^* glomus cells (Figure 4-figure supplement 1D), consistent with a defect in ψM in normoxia in *Higd1c^-/-^* glomus cells.

In developing our Rh123 imaging protocol, we tested several Rh123 concentrations up to 50 µg/ml. We did not want to increase the concentration further because Rh123 and other mitochondrial membrane potential dyes (TMRM and TMRE) begin to inhibit mitochondrial respiration at this concentration (Scaduto and Grotyohann, 1999, Biophys J). We also found empirically that extending the incubation time for dye loading did not improve the FCCP response.

18. If you want to see if genotype affects resting ψM you would probably be better not working in the quench mode but either using much lower concentrations of Rh123 or another probe, e.g. TMRM, and loading for a longer period of time (until dye uptake reaches a steady state without any quenching occuring).

To interpret imaging experiments with Rh123 and TMRM in non-quenching mode to measure resting ψM would require perfectly controlled and consistent dye loading and imaging conditions in order to compare absolute fluorescence differences. As far as we know, this has not been done in glomus cells, even using dissociated cell preparations. When performing two-photon imaging of intact carotid bodies, glomus cells are located at different depths and necessarily experience different illuminations, along with differences in exposure to Rh123 during loading. Therefore, we do not believe we can achieve the level of control necessary to draw a strong conclusion using Rh123 or TMRM in the non-quenching mode in intact tissue. We believe that testing FCCP responses by Rh123 imaging in quenching mode provides a reasonable assessment of the maximal response, especially if we deliver the stimulus at the beginning of the experiment when cells across preparations are likely to be in the most similar state (see responses to items 15-17).

19. Data in Fig6-E shows >= 40% inhibition of oxygen consumption under hypoxic conditions compared to normoxia in all HEK cell types studied. Given that the level of oxygen in hypoxia is stated to be 25 mmHg this is something of a surprise. The P50 for most cells is thought to be < 1mmHg. Something is wrong here. If you are using an Oroboros Oxygraph (as stated in methods) it should be possible to measure oxygen consumption from air all the way down to zero oxygen and derive an exact P50 for each cell type. This should be done. This is a very important experiment that proports to show that it is the combination of Higd1c and Cox4I2 that generates the unusual sensitivity of complex IV towards oxygen. Philosophically I like this hypothesis. If true it would mark a major advance in this field, but I want to see some hard data with rigorously determined kinetic measurements!

Thank you for this suggestion. We agree with the importance of deriving p50 values for *HIGD1A-KO* cells expressing different combinations of HIGD1C and COX4I2 to extend our data in Figure 6E and F. We performed additional experiments on these cell lines to calculate the P50 using an Oroboros Oxygraph as suggested. The new data shows that co-expression of HIGD1C and COX4I2 in *HIGD1A*-KO cells increased the mitochondrial oxygen affinity (p50_mito_) to reveal an enhanced response to hypoxia (p. 37, new Figure 6F and Figure 6—figure supplement 1E). Notice that in these assays, to calculate the p50_mito_, we measured endogenous whole cell respiration.

20. It would be better in future if the calcium imaging studies could be performed using the same perfused preparation that the nerve recording experiments used. This would remove the requirement for equilibrating superfusate with 95% oxygen which is far from physiological and would allow the carotid body tissue to be perfused with physiologically relevant levels of oxygen. However, superfusion with hyperoxic solutions is considered standard in many systems and so additional experiments are not required at this time.

We thank the reviewer for this suggestion, which we will strive to adopt in future calcium imaging experiments.

21. Expanding the study to see if HIGD1A is expressed in carotid bodies of multiple species would strengthen the paper. It would also be interesting to see if it is absent in the carotid bodies of guinea pigs which are not acutely oxygen sensitive. Furthermore, chromaffin cells in fetal adrenal medulla are oxygen-sensitive whereas mature chromaffin cells are not, it would be fascinating to see if HIGD1A expression changes during the maturation of chromaffin cells. These possibilities might be discussed.

We showed that *Higd1c* is expressed in the carotid bodies of both mouse and human. We appreciate the suggestion to expand the study to additional species, such as the poorly oxygen-sensitive guinea pig. We have added RT-qPCR data from the rat, another common animal model for carotid body studies. *Higd1c* was expressed at higher levels in the carotid body compared to adult and neonatal adrenal medulla and thoracic spinal cord, which contains a central oxygen sensor (Barioni et al., 2022, Sci Advances) (p. 6, Figure 1-figure supplement 4B). *Higd1c* expression in the adrenal medulla did not increase from neonate to adult. Testing expression in other species is beyond our capabilities at this time because of limited access to carotid body tissue.

22. Tidal volume changes with the weight of the animals. Tidal volume should be adjusted for the weights of the animals tested. If there are weight differences between knockouts and wildtype it can have profound effects on the data.

Body weights of *Higd1c^+/+^* and *Higd1c^-/-^* animals for all three *Higd1c* alleles were not significantly different by two-way ANOVA with Sidak correction (see Author response image 2).

We did consider normalizing tidal volume and minute ventilation to body weight as suggested, but for the animals used in our study, body weight did not correlate well with respiratory rate, tidal volume, or minute ventilation in wild-type or mutant animals in normoxia or hypoxia. When plotted against body weight, only tidal volumes for *3-1^+/+^* in normoxia and *5-3^+/+^* in hypoxia had fitted regression lines with significantly non-zero slopes at p<0.05. Nevertheless, we verified that all differences that are statistically significant (p<0.05) without normalizing by body weight are also significant if we normalize to body weight. In addition to comparing mean values of ventilatory parameters, we showed the % change in the ventilatory parameters for each animal before and after hypoxia (“Hypoxic Response”), which we feel is the most appropriate presentation as it essentially uses each animal as its own control. We have added our rationale for not normalizing by body weight in the Methods (pp. 47-48).

Reviewer #1 (Recommendations for the authors):Understanding the precise intracellular signalling pathways underpinning hypoxic sensitivity in the carotid body chemoreceptor has been a major unsolved area in sensory neurophysiology. Early studies almost 60 years ago suggested the importance of a metabolic signal coupled to the mitchrondria, which could regulate intracellular calcium homeostasis and exocystosis of excitatory transmitter in type I glomus cells.Using RNAseq this manuscript has identified a novel gene that encodes protein regulation of Complex IV in the carotid body response to hypoxia. Here the authors found HIGD1C, a novel hypoxia-inducible gene domain factor isoform, as an electron transport chain Complex IV-interacting protein was almost exclusively expressed in the carotid body. Using a combination molecular biology, protein chemistry, genetic knock-out, calcium imaging and respiratory measurements, the authors elegantly demonstrated the physiological utlility of HIGD1C being required for carotid body oxygen sensing via enhanced Complex IV sensitivity to hypoxia. Deletion of the gene massively attenuated the chemoreceptor response to hypoxia, whereas of expression of HIGD1C could recapitulate the increased oxygen sensitivity of Complex IV in HEK 293T cells, but not the response to hypercapnia.Whilst this large body of work shows the putative role of HIGD1C in oxygen sensing, the relationship between this protein and the graded response of afferent nerve firing to physiological levels of hypoxia remains to be established to support an essential role for this signalling pathway in hypoxic chemotransduction.This is a large body of work supporting the idea that HIGD1C is required for hypoxic sensing in the mouse through the regulation complex IV. The experiments are well designed, complex and appear to have been carefully performed.

We thank the reviewer for recognizing the quality of our study.

Specific comments for improvement:Parts of the manuscript are difficult to follow re data presentation since the data are densely reported in places. Eg Figure 1 Figure supplement 2 on page 9 is virtually impossible to read.

Please see response to item 1.

A tighter introduction is required re references where credit should be given to the early studies showing that hypoxia excites CB and breathing and chemodenervation abolishes the breathing responses (eg Neil; Torrance; Lahiri should be given credit in para 1 on page 3).

Please see response to item 2.

Additionally, primary references should be given for statements of fact like …'Under these pathological conditions, suppressing CB activity improves causal symptoms such as hypertension (REF), cardiac arrhythmias (REF), and insulin resistance (REF) (Iturriaga, 2018)'. Credit must be given to the primary source, not a review.

Please see response to item 3.

p 4 'We found that three such genes, Higd1c, Cox4i2, and Ndufa4l2, were expressed at higher levels in the mouse CB (Figure 1A)'. Are you confident that this expression is only in type 1 cells and not type 2 cells?

Please see response to item 4.

Much of the data presented has used parametric statistics on n of 3/3 or 4/4 samples eg Figure 1 supp 3 page 10. Are these data normally distributed to support parametric use of stats? Please comment on the sample re biological replicates v technical replicates. How were the data treated here?

Please see response to item 5.

The gel on p26 panel B looks over run. Comment?

Please see response to item 6.

Aspects of the data are very convincing. Whilst this large body of work shows the putative role of HIGD1C in oxygen sensing, the relationship between this protein and the graded response of afferent nerve firing to physiological levels of hypoxia remains to be established to support an essential role for this signalling pathway in hypoxic chemotransduction. This limitation should be acknowledged. Please comment?

Please see response to item 7.

Reviewer #2 (Recommendations for the authors):I have a number of major concerns about parts of this research which need to be addressed before I could comment on its conclusions and importance. Some of these may just require a better explanation or correction to the text (e.g. where there may be an ambiguity over how an experiment was conducted). Others may require further work

We thank the reviewer for detailed assessment of our manuscript. We have tried to provide additional clarification as requested and performed additional experiments.

Major comments/QuestionsGeneral reporting of resultsA very general comment I have about this paper is that numerous experiments, particularly in the latter half of this paper using HEK cells, have been conducted with an 'n' of only 3. I do not know what eLifes's normal expectations/requirements are but I would have thought that an 'n' of at least 4 or 5 independent observations should normally be required. Similarly, numerous gels/blots are presented without any indication of how often these types of experiments were repeated. This should be reported and the journal should have some policy over what is expected. To my mind there could be a substantial amount of further work that needs to be completed before this study could be formally published. Apart from this issue the statistics seem to have been done with suitable care and rigour.

For most quantitative data in cell culture, a minimum n=3 (independent replicates) is enough if there is not much variability, and we have tested for normality using the Shapiro-Wilk test to justify use of parametric statistical tests when appropriate. We increased the sample size for RT-PCR experiments to increase our confidence in those results. Please see also response to item 5 above.

All gels and blots were repeated 3 times, except for *in-gel* activity experiments in Figure 5-figure supplement 1E, which were performed twice. We added the number of replicates to figure legends as appropriate (Figure 5 and Figure 5-figure supplements 1-3). n values for data present in bar graphs were already reported in legends.

Questions/suggested changes.Plethysmography.With respect to the effects of Higd1c KO on ventilation. Although all KO's appear to have a blunted ventilatory response to hypoxia, a ventilatory response remains none the less. This is particularly evident for the 1.1 KO as shown in the example trace in Figure 2 supplement 1 A where the second response to a hypoxic stimulus looks very similar between the KO and normal mouse. This is a somewhat disappointing result in that it may limit the conclusions that can be drawn from this study. One could have hoped that a Higd1c KO would either eliminate the response to hypoxia or had no effect at all. As this is a very important piece of evidence I want to ask for more detail regarding the data and methods.

Please see response to item 8.

Forgive me if I have misunderstood but in the methods section you state that "we used a respiratory rate of 215 breaths/min (comparable to the mean of a wild type in hypoxia) as a cut off for inclusion in this study". It is unclear how this works. Are you rejecting recordings with RR in excess of this figure or below this figure? When are you making this decision, under normoxic conditions at the beginning of an experiment, under hypoxic conditions, or at any time in the recording? Can you clarify this point and indicate how often recordings were rejected by this criteria in the different mice?

Please see response to item 9.

Was there any notable difference in mouse behaviour between genotypes during plethysmography? Mice usually become much less active in hypoxia and basal metabolic rate goes down resulting in a fall in CO2 production and a drop in PaCO2 which leads to a fall in ventilation (hypoxic ventilatory decline: HVD). This often complicates the analysis/interpretation of responses to hypoxia. Over what time period do you measure changes in ventilation – over the entire period of exposure to hypoxia, at the peak? Or towards the end? Are changes in the rate of HVD influencing the comparison between KO and wt? Have you measured PaCO2 under hypoxic conditions? I would predict that it would be lower than under control conditions. Could you introduce a little CO2 in hypoxic conditions to try to neutralise this effect?

Please see response to item 10.

What happens if you do a sham switch (i.e. switch between two identical gasses/air), is there any startle response associated with just mechanically switching between two gas lines? If not what happens if you look specifically at the peak/early response to hypoxia before CO2 has much chance to fall?

Please see response to item 11.

Are Higd1c +/+ mice taken from your in house breeding programs generating Higd1c -/- mice or are they just off the shelf C57Bl/6J from Jax? If the latter how long are these mice held in your facility (under the same conditions as the KO mice) before being used?

Please see response to item 12.

Nerve fibre recordings and calcium recordings.In the methods section on nerve recordings you state that "Preparations were exposed to a brief hypoxic challenge (PO2 = 60 mmHg) to determine viability; preparations that failed to show a clear increase in activity during this challenge were discarded". You are testing the hypothesis that Higd1c is important/essential for the generation of the response to hypoxia so why exclude data from preparations that lack a hypoxic response? This is just what you are looking for! I hope this is just a mistake in writing up the methods, BUT if this is really what was done then the whole nerve recording study could be invalid (depending on just what proportion of preparations were actually discarded). If you want some sort of test for viability of the preparation, which is a good idea, then there are many other stimuli one could choose instead that do not involve mitochondrial function e.g hypercapnia/acidosis, high potassium, acetyl choline, TASK channel inhibitors/respiratory stimulants.

Please see response to item 13.

One of the key issues regarding the role of alternative mitochondrial subunits in the carotid body is not simply that they may affect the maximal turnover rate of cytochrome oxidase but whether they alter the kinetic parameters defining sensitivity to hypoxia (i.e. Km, oxygen affinity, P50 ). To determine these really requires making a number of measurements at different PO2 including a zero point (anoxia). This could have been done for both nerve recordings and calcium measurements. Unfortunately it was not, so we are left with data from both of these experiments which demonstrates that the response to hypoxia is smaller in the KO but we have no idea whether these kinetic parameters have changed or not. The study would be a lot stronger if you could demonstrate a change in Km in the Higd1c KO.

Please see response to item 14.

Rh123 experiments.It would appear from the results obtained with Rh123 and the weak responses to FCCP that there was inadequate loading of this dye to make good measurements of mitochondrial potential (ψM) in many of the cells studied. If you look at the paper by Biscoe and Duchen they were achieving a 100% increase in Rh125 fluorescence upon adding FCCP, Anoxia or CN, not 20%. The level of loading is critical with this dye in order to get it to work properly in the de-quench mode. In addition, to correct for differences in dye loading, responses to acute hypoxia should probably be normalised not only to the baseline but also to the response to FCCP. The smaller response to hypoxia in Higd1c -/- could simply reflect lower dye loading, with consequent lower dynamic response from Rh123 to change in mitochondrial potential, rather than any actual change in ψM.

Please see response to item 15.

The response to anoxia has not been tested which again confounds attempts to determine a P50 for mitochondrial depolarisation. I note that the rate of change in PO2 (Figure 4 suppl 1 A) is rather slow so some Rh123 may also leak out of the cell during the recording. Were any corrections applied for dye leakage?

Please see response to item 16.

In summary the interpretation of this data is equivocal. Higd1c -/- may take up Rh123 less avidly than wt but is this due to a lower resting ψM or something else? This experiment needs repeating extending loading time or Rh123 concentration until robust responses to FCCP can be recorded.

Please see response to item 17.

If you want to see if genotype affects resting ψM you would probably be better not working in the quench mode but either using much lower concentrations of Rh123 or another probe, e.g. TMRM, and loading for a longer period of time (until dye uptake reaches a steady state without any quenching occuring).

Please see response to item 18.

CIV sensitivity to Hypoxia in HEK.Data in Fig6-E shows >= 40% inhibition of oxygen consumption under hypoxic conditions compared to normoxia in all HEK cell types studied. Given that the level of oxygen in hypoxia is stated to be 25 mmHg this is something of a surprise. The P50 for most cells is thought to be < 1mmHg. Something is wrong here. If you are using an Oroboros Oxygraph (as stated in methods) it should be possible to measure oxygen consumption from air all the way down to zero oxygen and derive an exact P50 for each cell type. This should be done. This is a very important experiment that proports to show that it is the combination of Higd1c and Cox4I2 that generates the unusual sensitivity of complex IV towards oxygen. Philosophically I like this hypothesis. If true it would mark a major advance in this field, but I want to see some hard data with rigorously determined kinetic measurements!

Please see response to item 19.

Reviewer #3 (Recommendations for the authors):This paper uses a range of techniques to demonstrate the presence of a novel protein (HIGD1C) that interacts with complex 4 of the mitochondrial electron transport chain. The authors demonstrate that this protein is required for hypoxic chemotransduction at the level of the whole animal (plethysmography), at the level of the in-vitro organ (carotid sinus nerve recordings) and at the level of the glomus cell (calcium imaging). The authors then go on to demonstrate that HIGD1C may interact and alter the sensitivity of complex 4 to hypoxia.The authors are to be congratulated on a set of thorough physiological experiments that are then extended by detailed cellular respiration studies. Working with mouse carotid body tissue is incredibly challenging and the authors have done extremely well to get such high quality and valuable data.The combination of genetic, physiological and cellular experiments make this an extremely compelling paper suitable for publication in eLife.The question of why carotid body glomus cells are unusually sensitive to hypoxia has troubled researchers for decades. This paper makes an extremely compelling case for the expression of HIGD1C modulating complex 4 and thereby sensitizing it to hypoxia. The data seems strong and the paper is likely to have a major impact in the field of oxygen sensing.

We appreciate the reviewer’s recognition of the challenges of performing carotid body experiments in mice, the quality of our data, and the impact of our findings to the field. Thank you!

There are several observations that would greatly strengthen the paper.1. It would be better in future if the calcium imaging studies could be performed using the same perfused preparation that the nerve recording experiments used. This would remove the requirement for equilibrating superfusate with 95% oxygen which is far from physiological and would allow the carotid body tissue to be perfused with physiologically relevant levels of oxygen. However, superfusion with hyperoxic solutions is considered standard in many systems and so additional experiments are not required at this time.

Please see response to item 20.

2. Expanding the study to see if HIGD1C is expressed in carotid bodies of multiple species would strengthen the paper. It would also be interesting to see if it is absent in the carotid bodies of guinea pigs which are not acutely oxygen sensitive. Furthermore, chromaffin cells in fetal adrenal medulla are oxygen-sensitive whereas mature chromaffin cells are not, it would be fascinating to see if HIGD1C expression changes during the maturation of chromaffin cells. These possibilities might be discussed.

Please see response to item 21.

3. Tidal volume changes with the weight of the animals. Tidal volume should be adjusted for the weights of the animals tested. If there are weight differences between knockouts and wildtype it can have profound effects on the data.

Please see response to item 22.